

# Secondary aerosol formation promotes water uptake
# by organic-rich wildfire haze particles in Equatorial
# Asia
**Jing Chen[1, *], Sri Hapsari Budisulistiorini[1], Takuma Miyakawa[2], Yuichi Komazaki[2],**
**Mikinori Kuwata[1, 3, 4, *]**
[1] {Earth Observatory of Singapore, Nanyang Technological University, Singapore, Singapore}
[2] {Research and Development Center for Global Change, Japan Agency for Marine-Earth
Science and Technology, Yokosuka, Japan}
[3] {Asian School of Environment, Nanyang Technological University, Singapore, Singapore}
[4] {Campus for Research Excellence and Technological Enterprise (CREATE) program,
Singapore, Singapore}
* Correspondence to: chen.jing@ntu.edu.sg; kuwata@ntu.edu.sg



# 1 Abstract

Diameter growth factors ($GF$) of 100 nm haze particles at 85 % relative humidity and chemical characteristics were simultaneously monitored at Singapore in October 2015 during a pervasive wildfire haze episode, which was caused by peatland burning in Indonesia. Non-refractory submicron particles ($NR-PM_1$) were dominated by organics (approximating 77.1 % in total mass), whereas sulfate was the most abundant inorganic constituent (11.7 % on average). A statistical analysis of the organic mass spectra showed that most of organics (36.0 % of $NR-PM_1$ mass) were highly oxygenated. Diurnal variations of $GF$, number fraction of highly hygroscopic mode particles, mass fraction of sulfate, and mass fraction of oxygenated organics (OOA) synchronized well, peaking during daytime. The mean hygroscopicity parameter ($\kappa$) of haze particles was 0.189 ± 0.087, and mean $\kappa$ values of organics were 0.157 ± 0.108 ($\kappa_{org}$, bulk organics) and 0.287 ± 0.193 ($\kappa_{OOA}$, OOA), demonstrating the important roles of both sulfate and highly oxygenated organics in hygroscopic growth of wildfire haze particles. $\kappa_{org}$ was also affected by the water-soluble organic fraction to some extent. These results show the importance of secondary formation processes in promoting water uptake properties of wildfire haze particles, including both inorganic and organic species. Further detailed size-resolved as well as molecular level chemical information of organics will be necessary for more profound exploration of water uptake by wildfire haze particles in Equatorial Asia.





## 1. Introduction

In the last few decades, wildfire haze has been periodically raging equatorial Asian countries (Page et al., 2002; van der Werf et al., 2010; Field et al., 2016; Koplitz et al., 2016), resulting in billions of dollars of economic losses as well as thousands of premature deaths (Johnston et al., 2012; Marlier et al., 2013). The increasing wildfire activity is associated with recent development of tropical peatland (Page et al., 2009; Spracklen et al., 2015). Water table level of tropical peatland is artificially decreased by developing canals, enhancing flammability of peat (Langner et al., 2007; Konecny et al., 2016). The occurrence of peatland fire is closely related with El Niño-induced droughts (Page et al., 2002; Field et al., 2016). Enhanced peatland fire has been observed during intense El Niño years, including 1997, 2006, and 2015 (Page et al., 2002; van der Werf et al., 2010; Stockwell et al., 2016). The peatland fire in 1997 was globally important, as the total carbon emission was estimated to equal 13−40 % of the year's annual global carbon emission from fossil fuels (Page et al., 2002). The recent 2015 wildfire haze event could rival the one in 1997 in terms of not only hazardous to human health but also the significant impacts on global climate (Crippa et al, 2016; Field et al., 2016; Huijnen et al., 2016; Koplitz et al., 2016; Stockwell et al., 2016). In fact, if there were a prize for the worst air pollution disasters of the century, the 2015 equatorial Asian haze event would likely be nominated (Crippa et al, 2016; Stockwell et al., 2016). During September − October 2015, the thick smoke stemmed from peatland fires blanketed Equatorial Asia and released huge amounts of organic material and fine particulate matter (particulate matters with the aerodynamic diameter below 2.5 μm, $PM_{2.5}$) (Crippa et al, 2016; Koplitz et al., 2016), the leading cause of global air-pollution-related mortality (Kunii et al., 2002; World Health Organization, 2009; Johnston et al., 2012; Marlier et al., 2013; Lelieveld et al., 2015).

A previous study on the peatland fire event in 1997 has reported that the wildfire haze particles resulted in dramatically cooling effects on the atmosphere radiative budget, especially over the source region of Indonesia ($-150$ W m$^{-2}$) and the tropical Indian Ocean ($-10$ W m$^{-2}$) (Duncan et al., 2003). Along with the subsequently affected shallow warm clouds and deep convection processes, the resultant abnormal rainfall in adjacent tropical region and extra-tropics was also confirmed by both satellite observations and model simulations (Rosenfeld, 1999). These studies demonstrate the importance of investigating aerosol-cloud-precipitation



interactions of Indonesian wildfire haze particles, including water uptake property of aerosol
particles.
Previous studies on water uptake properties of aerosol particles originating from
Indonesian peatland fire are controversial. Laboratory studies have demonstrated that fresh
Indonesian peat burning particles are weakly hygroscopic and almost inactive as cloud
condensation nuclei (CCN) (Chand et al., 2005; Dusek et al., 2005; Chen et al., 2017). On the
other hand, a field observation showed that the wildfire haze particles were highly hygroscopic
during the 1997 Indonesian peatland fires (Gras et al., 1999). The discrepancy impedes further
reliable evaluations on regional and global climate impacts driven by Indonesian wildfire haze
particles (Lin et al., 2013; Reid et al., 2013). The cause of the discrepancy will need to be
quantitatively understood in comparison with particle chemical composition.
Aerosol particles emitted from wildfire fire is a mixture of both inorganic and organic
compounds, complicating their water uptake properties (Carrico et al., 2008, 2010; Petters et al.,
2009; Hallar et al., 2013; Lathem et al., 2013). Water uptake properties of inorganic salts such as
ammonium sulfate and ammonium nitrate are well known, yet hygroscopic behavior of organic
compounds or organic-inorganic mixtures are still difficult to predict due to the complex
chemical composition of organics and associated distinct affinity for water of a specific chemical
constitute (Saxena et al., 1995; Gysel et al., 2004; Dinar et al., 2007; Petters and Kreidenweis,
2007; Carrico et al., 2010; Kristensen et al., 2012; Marsh et al., 2017). For instance, experimental
and modelling studies have shown that water uptake by water-soluble matter is governed by the
inorganic fraction, whereas hygroscopic properties of inorganics can be altered substantially by
the presence of organics (Saxena et al., 1995; Dick et al., 2000). In general, with the increase of
organic fraction, water uptake by wildfire particles has performed an overall decreasing trend
(Mircea et al., 2005; Carrico et al., 2010), evidencing the high sensitivity of particle water uptake
to the organic fraction. The high sensitivity to organic fractions has been observed, as inorganic
species are much more hygroscopic than most of organic compounds. However, the roles of
inorganic and organic species in water uptake by Indonesian peatland burning particles have
rarely been investigated (Dusek et al., 2005; Chen et al., 2017).



Water uptake properties of organic compounds have also been demonstrated to be
important, especially when chemical composition of aerosol particles is dominated by organics.
Such cases are frequently observed for particles emitted from wildfires (Petters et al., 2009;
Carrico et al., 2010; Cubison et al., 2011; Hallar et al., 2013; Chen et al., 2017). Both theoretical
and experimental studies demonstrated that water-soluble organic matter (WSOM) plays the key
role in determining water uptake by organic compounds (Peng et al., 2001; Gysel et al., 2004;
Petters and Kreidenweis, 2007; Carrico et al., 2008; Petters et al., 2009; Lathem et al., 2013;
Chen et al., 2017). For instance, freshly emitted peat burning particles are known to contain a
small fraction of WSOM, explaining their limited hygroscopicity (Chen et al., 2017). Chemical
aging and oxidation of organic compounds both in gas and particle phases could alter water
uptake properties of aerosol particles in a wildfire plume (Gras et al., 1999; Petters et al., 2009;
Rose et al., 2010; Cubison et al., 2011). These chemical processes in the atmosphere enhance
fractions of highly oxygenated organics and polar species, which are typically water soluble
(Duplissy et al., 2008, 2011; Jimenez et al., 2009; Chang et al., 2010; Massoli et al., 2010;
Cubison et al., 2011; Cerully et al., 2015). Although such chemical transformation and
corresponding changes in hygroscopicity have been observed for wildfire particles both in
laboratory and field (Petters et al., 2009; Massoli et al., 2010; Rose et al., 2010; Cubison et al.,
2011; Duplissy et al., 2011), the importance of these processes on water uptake property has
never been investigated for peatland burning particles in equatorial Asian region.
In this work, we investigated the relationships between water uptake properties and
chemical composition of aerosol particles in tropical peatland fire haze in October 2015 by
conducting atmospheric observation in Singapore. We quantified water uptake properties using
the Humidified Tandem Differential Mobility Analyzer (HTDMA). In parallel, particle chemical
composition was characterized in real-time using the Aerodyne Time of Flight-Aerosol Chemical
Speciation Monitor (ToF-ACSM). Further, water-soluble organic carbon (WSOC) and elemental
carbon (EC) contents were quantified with off-line analysis using ambient $PM_{2.5}$ filter samples.
The data from these measurements were combined to explore how water uptake property of
tropical peatland burning particles is regulated.



**2. Observation**
**2.1. Field Campaign**
The field observation was conducted at the campus of Nanyang Technological University
(NTU), Singapore (1°20′41″ N, 103°40′53″ E) during October 2015. The campus is located at
20 km away from the city center, and surrounded by a secondary tropical forest and grassland.
The site is located 0.8 km away from a highway, and a petrochemical complex (Jurong Island) is
located approximately around 8 km south.
The observation was performed in an air-conditioned room, with the room temperature
kept around 22 °C. A cyclone (URG-2000-30EN PM$_{2.5}$, URG) was employed for ambient
aerosol sampling at a flow rate of 16.67 L min$^{-1}$. The inlet was fixed on the rooftop, which is
located approximately 10 m above the ground. The sample air was split into several flows for
different instruments after drying by diffusion dryers (with the relative humidity, $RH$, of the
sample flow below 30 %). During the observation, particle number size distribution, chemical
composition, and hygroscopic growth were monitored.
**2.2. Particle water uptake measurements**
Water uptake property was measured using the HTDMA system (Chen et al., 2017).
Briefly, sampled particles were desiccated using a diffusion dryer (Model 42000, Brechtel
Manufacturing, Inc.), and resulting dry polydisperse particles were classified by the first
differential mobility analyzer (DMA, Model 3081, TSI Inc.). The DMA selects particles of a
specific mobility diameter ($D_0$), which was fixed at 100 nm during the observation. The
classified particles were humidified to $RH$ = 85 % using nafion tubings (MD-110-12S-4, Perma
Pure) operated under a controlled $RH$ condition. The particle residence time in the humidifier
was approximately 10 seconds. The variation of $RH$ was ± 0.5 % (peak to peak). The resulting
size distribution of humidified particles was measured by the second DMA coupled with a
condensation particle counter (CPC, Model 3775, TSI Inc.). The diameter growth factor
parameter, $g$, which is defined as the ratio of the particle diameter after humidification at a
conditioned $RH$ ($D_p(RH)$) to the initial dry size ($D_0$) (i.e., $g = D_P(RH)/D_0$ ), was calculated from





the HTDMA data. Hygroscopicity parameter, $\kappa$, can be derived from the corresponding $g$ at a
given $RH$ and $D_0$ using the following equation (Petters and Kreidenweis, 2007):

$$\kappa = (g^3 - 1) \cdot \left( \frac{1}{RH} \cdot \exp\left( \frac{4\sigma_{s/a} \cdot M_w}{\rho_w \cdot R \cdot T \cdot D_0 \cdot g} \right) - 1 \right),$$
(1)

where $\sigma_{s/a}$ is the surface tension of the solution/air interface (0.0718 N m$^{-1}$ at 25 °C), $M_w$ and $\rho_w$
are the molecular weight (0.018 kg mol$^{-1}$) and density of water ($1 \times 10^3$ kg m$^{-3}$), respectively, $R$
is the universal gas constant (8.31 J K$^{-1}$ mol$^{-1}$), and $T$ is absolute temperature (298 K). Further
details about the HTDMA are available in Chen et al. (2017).
**2.3. Aerosol chemical analysis**

9        The ToF-ACSM (Aerodyne Inc.) measured the chemical composition of non-refractory

submicron particles (NR-PM$_1$), including organics (OA), sulfate (SO$_4^{2-}$), nitrate (NO$_3^-$),
ammonium (NH$_4^+$), and chloride (Cl$^-$) (Fröhlich et al., 2013). The ToF-ACSM sampled particles
desiccated by a nafion tubing. The organic mass spectra measured by the ToF-ACSM were
analyzed in detail using a Multilinear Engine (ME-2 solver) software (Canonaco et al., 2013).
Four specific OA types were identified, i.e., hydrocarbon-like OA (HOA), peat burning OA
(PBOA), non-peat biomass burning OA (briefly, BBOA), and oxygenated OA (OOA). Details
about the ToF-ACSM measurements and data analysis are provided in Budisulistiorini et al. (in
preparation).

18       PM$_{2.5}$ filter samples for chemical analysis were also collected using filter holders (BGI

Inc.). The samples were collected for 24 hours using 47 mm (diameter) quartz-fiber filters. The
sampling started/ended at 08:00 local time (LT). The collected samples were analyzed for bulk
OC, EC, and WSOC. All the quartz-fiber filters were prebaked at 900 ºC for 3 hours before
sampling; after sampling, they were stored in a refrigerator (−20 ℃) until analysis. For each
sampling, a back-up quartz-fiber filter was used to account for potential influence of gas phase
organic components on the particulate organics collected on the front quartz-fiber filter (Turpin
et al., 1994). The method assumes that all the particulate OC is collected by the front filter, while
gas phase OC is equally collected on both front and back filters. Subtraction of the OC loading



on the back filter (i.e., gas phase OC) from that on the front one allows quantification of
particulate OC (i.e., corrected OC).
Concentrations of OC and EC were determined by thermal-optical reflectance analysis
(Chow et al., 1993) using the Sunset Laboratory OC/EC Analyzer following the IMPROVE-A
protocol. WSOC was quantified with the Sievers 800 Total Organic Carbon (TOC) Analyzer
following extraction of a part of filter sample (8 mm φ) by 10 ml of HPLC-grade water. An
orbital shaker was operated for 21 hours for the extraction, and the subsequent solutions were
filtered with syringe filters (pore size of 0.2 μm).
**2.4. Particle number size distribution**
Particle number size distributions were measured using a NanoScan SMPS Nanoparticle
Sizer (NanoScan-SMPS, Model 3910, TSI Inc.) and an Optical Particle Sizer (OPS, Model 3330,
TSI Inc.). The detected particle size ranges are $11.5 − 365.2$ nm (NanoScan-SMPS) in mobility
size and $0.3 − 10$ μm (OPS) in aerodynamic size. Both the instruments sampled particles
desiccated by a diffusion dryer (Model 42000, Brechtel Manufacturing, Inc.). Time resolutions
of both the instruments were 1 minute.
**3. HTDMA data analysis**
**3.1. Classification of three hygroscopic modes**
Figure 1 shows the HTDMA data averaged over the whole observation period. The mean
normalized particle size distribution after humidification at 85 % *RH* has spanned a few different
modes, reflecting mixing states of ambient wildfire haze particles observed at Singapore
(Bougiatioti et al., 2016; Ogawa et al., 2016). For particles in a specific hygroscopic mode, *i*,
with $g_{1,i} < GF < g_{2,i}$, the number fraction of this mode ($nf_i$) can be derived from measured
probability density function of $g$ (i.e., $c(g, D_0)$) as $nf_i = \int_{g_{1,i}}^{g_{2,i}} c(g, D_0)\, dg$ . The corresponding





mean *GF* ($g_{mean,i}$) is calculated from $g_{mean,i} = \dfrac{1}{nf_i} \int_{g_{1,i}}^{g_{2,i}} g\ c(g, D_0)\ dg$ (Gysel et al., 2009). The
equivalent values of $\kappa$ for mode $i$ is obtained from $g_{mean,i}$ using Eq. (1).

3        Observed 100 nm dry particles were categorized into the following three groups based on

their hygroscopic properties at *RH* = 85 %, facilitating analysis of heterogeneity of particle
chemical composition.
(1) Nearly non-hygroscopic or weakly hygroscopic particles ($0 \leq \kappa < 0.1$; $g < 1.15$), those are
dominantly composed of black carbon (BC), non-polar hydrocarbon-like organic compounds
(Peng et al., 2001; Gysel et al., 2007; Kreidenweis et al., 2008);
(2) Moderately hygroscopic particles ($0.1 \leq \kappa < 0.2$; $1.15 \leq g < 1.27$), which could contain
hygroscopic organics (e.g., carboxylic acids and levoglucosan) and/or mixtures of non/less and
more hydrophilic compounds (e.g., BC, fatty acids, and/or humic-like substances mixed with
ammonium sulfate or levoglucosan-like species) (Peng et al., 2001; Chan and Chan., 2003; Gysel
et al., 2004, 2007; Chan et al., 2005; Petters and Kreidenweis, 2007);
(3) Highly hygroscopic particles ($\kappa \geq 0.2$; $g \geq 1.27$), those contain inorganic salts as well as some
highly hygroscopic organic species such as multifunctional organic acids (Peng et al., 2001;
Carrico et al., 2008; Duplissy et al., 2011; Ogawa et al., 2016).

17        In addition, a volume-weighted mean growth factor, *GF*, was also calculated using $c(g,$

$D_0)$ (Gysel et al., 2009):
$$GF = \left( \int_0^{\infty} g^3 \cdot c(g, D_0)\ dg \right)^{1/3}. \tag{2}$$
*GF* was employed for calculating mean values of $\kappa$, facilitating comparison with chemical
composition of aerosol particles.
**3.2. Effective $\kappa$ of organic compounds ($\kappa_{org}$)**

23        Water uptake properties of organic compounds were estimated using the Zdanovskii-

Stokes-Robinson (ZSR) mixing rule, employing observed values of $\kappa$ and chemical composition




as input parameters. The ZSR mixing rule assumes that water uptake by a mixture of materials is
additive of water content retained by each chemical species (Stokes and Robinson, 1966). The
rule also assumes that the volume change of the mixing of species within individual particles is
almost negligible (Brechtel and Kreidenweis, 2000, Gysel et al., 2007; Petters and Kreidenweis,

5   2007):

$$\kappa = \sum_i \kappa_i \cdot \varepsilon_i = \kappa_{SNA} \cdot \varepsilon_{SNA} + \kappa_{org} \cdot \varepsilon_{org} + \kappa_{EC} \cdot \varepsilon_{EC}$$
$$\Leftrightarrow \kappa_{org} = \frac{\kappa - \kappa_{SNA} \cdot \varepsilon_{SNA} - \kappa_{EC} \cdot \varepsilon_{EC}}{\varepsilon_{org}} \qquad , \tag{3}$$

where $\kappa_i$ and $\varepsilon_i$ stand for the hygroscopicity parameter and volume fraction of a specific
component $i$ in dry particles, respectively. The subscript $SNA$ represents three major inorganic
salts including sulfate, nitrate, and ammonium; $org$ denotes organic species; $EC$ indicates
elemental carbon.

11       Sulfate, ammonium, and nitrate were considered for inorganics, the majority of which is

contributed by sulfate (see Table 2). Other materials such as crustal elements were neglected
because they are scarce for submicron wildfire haze particles in Southeast Asia
(Balasubramanian et al., 2003; Keywood et al., 2003; Stockwell et al., 2016). Both sulfate and
nitrate were almost completely neutralized by ammonia, and these three most abundant inorganic
constituents were combined together and assumed as ammonium sulfate (i.e., $\varepsilon_{SNA} = \varepsilon_{SO4} + \varepsilon_{NO3} +$
$\varepsilon_{NH4} \approx \varepsilon_{AS}$). Thus the value of $\kappa_{SNA}$ was considered as approximating the $\kappa$ value of ammonium
sulfate under the condition when sulfate dominates inorganics (Gunthe et al., 2009; Chang et al.,
2010; Ogawa et al., 2016). The elemental carbon (EC) is known as non-hygroscopic (i.e., $\kappa_{EC} \approx$

20   0).

21       Values of densities are required to compute $\varepsilon_i$ from observed mass fractions. The mass

fraction is taken as the first-order approximation of the volume fraction based on the hypothesis
that the bulk particle density is similar to the densities of individual compounds assuming
volume additivity (Kreidenweis et al., 2008; Gunthe et al., 2009; Hallquist et al., 2009). This
hypothesis is demonstrated to be acceptable when particles are composed primarily of organics
and sulfate (Cross et al., 2007; King et al., 2007). Densities of ammonium sulfate and EC were
assumed as 1.77 g cm$^{-3}$ and 1.80 g cm$^{-3}$, respectively (Park et al., 2004; Bond and Bergstrom,



2006). Density of organics is known to vary, depending on its elemental composition (Kuwata et
al., 2012). The value was assumed as 1.40 g cm$^{-3}$, which is a typical value for ambient organic
aerosols (Hallquist et al., 2009).
EC mass fraction of approximating 10.0 % in submicron wildfire haze particles was
simply utilized according to the time-averaged EC content in ambient PM$_{2.5}$ filter samples (see
Table 2). This assumption is on the basis of the preconditions that wildfire haze particles are
homogeneously mixed among different sizes and that there is no significant difference between
BC and EC.
**3.3. $\kappa$ of oxygenated organic compounds ($\kappa_{OOA}$)**
As described in Sect. 2.3, organics were numerically segregated to HOA, PBOA, BBOA,
and OOA. The value of $\kappa_{\mathrm{org}}$ can be calculated by a linear combination of contributions from
segregated fractions (Petters and Kreidenweis, 2007; Chang et al., 2010);
$$\kappa_{org} = \upsilon_{HOA} \cdot \kappa_{HOA} + \upsilon_{PBOA} \cdot \kappa_{PBOA} + \upsilon_{BBOA} \cdot \kappa_{BBOA} + \upsilon_{OOA} \cdot \kappa_{OOA}, \qquad (4)$$
where $\upsilon_i$ stands for the volume fraction of component $i$ in the whole organics.
Both hydrocarbon (-like) and peat burning OA are known to be almost non-hygroscopic,
meaning that their $\kappa$ values could be assumed as 0 (Gysel et al., 2007; Gunthe et al., 2009; Chang,
et al., 2010; Chen et al., 2017). Water uptake by freshly emitted biomass burning particles is
generally limited, especially compared with OOA (Carrico et al., 2010; Chang et al., 2010; Chen
et al., 2017). Water uptake by a mixed particle is largely driven by the relative abundance of
more and less hygroscopic components, and is more sensitive to uncertainties in hygroscopicity
of more hygroscopic compounds than that of less hygroscopic compounds (Gysel et al., 2007).
Thus, $\kappa_{\mathrm{BBOA}}$ was also assumed to be 0, considering that OOA always dominated over BBOA in
wildfire haze particles as suggested by our ToF-ACSM results (see Sect.4.2 and Fig. 6d). Under
these assumptions, $\kappa_{\mathrm{OOA}}$ can be calculated by the following equation:
$$\kappa_{OOA} = \kappa_{org} / \upsilon_{OOA}. \qquad (5)$$



Density of OOA is required to calculate $\upsilon_{OOA}$ using ME-2 resolved OOA mass
concentration combined with total volume concentration of bulk OA derived from ToF-ACSM
observed OA mass. The density of oxygenated organics was assumed to be 1.50 g cm$^{-3}$ (as
summarized in Table 1), which is appropriate typical value for carboxylic and multifunctional
organic acids (Saxena et al., 1995; Peng et al., 2001; Gysel et al., 2004; Carrico et al., 2010;
Ogawa et al., 2016). Detailed information on parameters utilized for the $\kappa_{org}$ calculation is
provided in Table 1.

**4. Results**

In this section, aerosol number size distribution (Figure 2), chemical composition
(Figures 3 and 4), and hygroscopic properties of aerosol particles (Figure 5) are shown, in
addition to diurnal variation of these data (Figure 6).

**4.1 Number size distribution of wildfire haze particles**

Figure 2a displays the time averaged particle size distribution within the whole size range
of 11.5 nm − 10 μm measured by NanoScan-SMPS combined with OPS. The NanoScan-SMPS
data were used for the fine particles (11.5 – 365.2 nm), and the overlapped size range of the OPS
(338 nm size bin) was excluded from the analysis. The data in the remaining OPS size range
(419 nm – 10 μm) were combined with the fine particle data. The temporal average size
distribution presents a unimodal structure with the number mode diameter located around
100 nm. Particles in the range of 30 – 200 nm dominate the total particle number concentration
whereas particles larger than 600 nm only account for a minor fraction (less than 4.0 % on
average), suggesting that wildfire haze particles in Singapore are predominantly contributed by
submicron particles.
Figure 2b shows the diurnal cycle of number size distribution. Particle number
concentrations (*i.e.*, dN/dlogD$_p$) higher than $1.5 \times 10^4$ cm$^{-3}$ are commonly observed in the 50 –
200 nm particle size range, while the number concentrations of super micron particles seldom
exceed $1.0 \times 10^3$ cm$^{-3}$. Particle number concentration was high in the morning of 08:00 –
09:00 LT. The concentration increased again after noontime (about 14:00 LT), lasting until





midnight. The high concentration periods could be caused by local traffic emissions and by
secondary formation. The diurnal variation was also observed for number concentration of whole
particles.

4        Figure 2c depicts the mean diurnal variations of the corresponding total particle number

and volume concentrations. The total number concentration started to increase after 07:00 LT
until around 10:00 LT, and reached the highest level after 14:00 LT. The particle number
concentration was higher than $1.5 \times 10^4$ cm$^{-3}$ before 19:00 LT. After that, the number
concentration decreased gradually, reaching $1.2 \times 10^4$ cm$^{-3}$ in the midnight. Correspondingly, the
aerosol volume concentration was higher than 50.0 µm$^3$ cm$^{-3}$ during daytime. The volume
concentration decreased during the nighttime, although it was still higher than 45.0 µm$^3$ cm$^{-3}$.
These results demonstrate that the aerosol loading was significantly high during the wildfire haze
pollution period.

**4.2 Chemical characteristics of wildfire haze particles**

14       Figures 3a-b show the times series of both mass concentrations and corresponding mass

fractions of organics, sulfate, nitrate, ammonium and chloride (expressed as $f_{org}$, $f_{SO4^{2-}}$, $f_{NO3^-}$,
$f_{NH4^+}$, and $f_{Cl^-}$, respectively, hereinafter) in NR-PM$_1$ quantified by the ToF-ACSM. The average
mass loading of NR-PM$_1$ was as high as $44.7 \pm 24.5$ µg m$^{-3}$, confirming the severity of the
pervasive wildfire haze. During the observation period, organics was always the most abundant
compound in NR-PM$_1$ ($34.8 \pm 20.7$ µg m$^{-3}$). The mass concentration of organics was higher than
50.0 µg m$^{-3}$ in many cases, and occasionally exceeded 100.0 µg m$^{-3}$. On average, organics
accounted for the highest mass fraction of 77.1 %, followed by sulfate (11.7 %), ammonium
(6.4 %), and nitrate (4.2 %). Mass concentration of chloride was almost negligible (0.6 % of the
total mass). These results demonstrate that submicron wildfire haze particles were predominantly
composed of organics.

25       Table 2 summarizes the mass concentrations of all the analyzed inorganic ionic species in

the PM$_{2.5}$ filter samples. The corresponding data for carbonaceous fractions are presented in
Fig. 4. On the whole, the mass fraction of EC varied from 4.4 % to 15.8 % with a mean value of
10.8 %. OC occupied $30.4 - 50.7$ % of the total PM$_{2.5}$ mass concentration, and the mean fraction
is 43.0 %. The WSOC fraction was in the range of $17.6 - 30.2$ % with a mean level of 26.7 %.





Correspondingly, the water-insoluble OC (WISOC) content was calculated to be $6.1 - 20.5\%$
with the mean fraction of 16.3 %. The WSOC/OC ratios were constantly higher than 50.0 % with
a mean and the maximum values of 63.6 % and 79.9 %, respectively. This result highlights that
the majority of organics in the wildfire haze particles were water soluble. Inorganic ions were
less abundant and variable than organics. On average, inorganics accounted for 30.5 % of the
$PM_{2.5}$ mass loading with the mean contributions of 0.6 % by $Cl^-$, 2.6 % by $NO_3^-$, 17.2 % by
$SO_4^{2-}$, 0.4 % by $Na^+$, 8.1 % by $NH_4^+$, 0.8 % by $K^+$, 0.1 % by $Mg^{2+}$, and 0.7 % by $Ca^{2+}$. Sulfate,
ammonium, and nitrate were the most abundant inorganic components. More than half of the
inorganics was contributed by sulfate. These results show that wildfire haze particles were
dominated by organics, especially water-soluble species. Mass concentrations of organics
measured by $PM_{2.5}$ filter samples and by the ToF-ACSM agreed well when organics/OC ratio
was assumed as 1.4 (slope = 1.07; $R^2$ = 0.91) (Reid et al., 2005; Hallquist et al., 2009; Levin et
al., 2010). The total mass concentrations of aerosol particles quantified by the filter samples and
the ToF-ACSM also correlated well ($R^2$ = 0.96). The mass loading of the $PM_{2.5}$ filter samples
was approximately 30 % higher than that of the ToF-ACSM results, likely because of the
difference in particle size range and lack of EC content for the ToF-ACSM measurements
(Budisulistiorini et al., in preparation).

18       Figure 3c shows the mean mass spectra of organics averaged over the observation period.

Ion signals at $m/z$ 43 (most likely $C_2H_3O^+$) and $m/z$ 44 ($CO_2^+$) were prominent, accounting for
7.5 % and 10.5 % of the total organics mass spectrum. The predominant signal of $m/z$ 44
indicates that organic compounds in wildfire haze particles were highly oxygenated. High-
molecular weight organic constituents with $m/z > 100$ possess an abundance of 13.3 % in total.
Marker ions for biomass burning particles such as $m/z$ 60 (mostly $C_2H_4O_2^+$) and $m/z$ 73 (mainly
$C_3H_5O_2^+$), those originating from levoglucosan-like species (e.g., levoglucosan, mannosan, and
galactosan) were also clearly observed (Cubison et al., 2011).

26       Figure 6d shows mean contributions of four organic components classified by the ME-2

method, including HOA (8.8 % of $NR-PM_1$ mass), PBOA (10.4 %), BBOA (10.0 %), and OOA
(36.0 %). Primary organics originating from biomass burning (i.e., PBOA and BBOA) accounted
for 20.4 % in total. OOA was the dominant type of organics during wildfire haze episodes.



### 4.3 Hygroscopic properties of wildfire haze particles

Figure 5 displays the time series of mean $GF$ data as well as corresponding values of $\kappa$ during the whole observation period. The mean values of $GF$ varied between 0.98 and 1.52 with the average of $1.25 \pm 0.09$. $GF$ values larger than 1.40 were normally observed after noontime. The variation of corresponding $\kappa$ results spanned from $0.004 - 0.475$, and the average of $\kappa$ value was $0.189 \pm 0.087$. Table 3 summarizes the mean $\kappa$ results of organics (cf. Sect.3.2, with EC considered) calculated from the HTDMA and ToF-ACSM measurements during the overlapping observation period of $10 - 24$ October 2015. The HTDMA-derived bulk $\kappa$ results averaged over the same overlapping period is defined as $\kappa_{HTDMA}$. The mean $\kappa_{org}$ ($0.157 \pm 0.108$) is lower than the mean $\kappa_{OOA}$ ($0.287 \pm 0.193$), as the whole organic fraction normally contains both non-hygroscopic and hygroscopic organics. The derived $\kappa_{org}$ and $\kappa_{OOA}$ results are demonstrated to be fairly comparable to previously reported $\kappa$ values for bulk organics and highly oxygenated water-soluble organic fraction, respectively (Petters and Kreidenweis, 2007; Chang et al., 2010; Duplissy et al., 2011; Lathem et al., 2013; Cerully et al., 2015; Ogawa et al., 2016; Chen et al., 2017). Note that the mean $\kappa_{OOA}$ is even higher than the corresponding mean $\kappa_{HTDMA}$ ($0.207 \pm 0.093$), revealing that the water uptake by the organic fraction particularly some highly oxygenated organics in the wildfire haze particles could be quite significant.

### 4.4 Diurnal variations of hygroscopic properties and chemical composition

$GF$ exhibits a clear pattern of diurnal variation (Fig. 6a). Higher $GF$ values were observed during the daytime ($GF = 1.27 \pm 0.05$ for $08:00 - 20:00$ LT). On the other hand, the value was lower in the early morning and nighttime ($1.23 \pm 0.05$ for $20:00 - 08:00$ LT). The corresponding mean bulk $\kappa$ results averaged over the whole observation period were $0.213 \pm 0.051$ for the daytime and $0.172 \pm 0.043$ for nighttime.

Similarly, the daytime mean $\kappa_{org}$ and $\kappa_{OOA}$ are $0.200 \pm 0.104$ and $0.353 \pm 0.179$, respectively, whereas the nighttime mean values are $0.103 \pm 0.086$ ($\kappa_{org}$) and $0.206 \pm 0.178$ ($\kappa_{OOA}$) (Table 3). These mean $\kappa$ values are 19.2 % lower ($\kappa_{org}$) and 42.8 % higher ($\kappa_{OOA}$) than the concurrently measured mean $\kappa_{HTDMA}$ result of $0.247 \pm 0.096$ (daytime), whilst 35.9 % lower ($\kappa_{org}$) and 28.2 % higher ($\kappa_{OOA}$) than that of $0.160 \pm 0.063$ (nighttime). A more significant discrepancy between $\kappa_{org}$ and $\kappa_{HTDMA}$ is observed for the nighttime case (whereas a larger difference between





$\kappa_{org}$ and $\kappa_{HTDMA}$ occurs during daytime), likely due to the greatly inhibited organics oxidation processes in the evening compared to the enhanced situation during daytime. Such a correlation could somewhat be visually clued from the corresponding diurnal patterns of both mean $GF$ and OA factors, as the fraction of OOA is demonstrated as a moderately good indicator of the hygroscopicity of organics (Ogawa et al., 2016).

The observed variation of $GF$ was dominantly caused by diurnal variation in probability distribution of $g$ (Fig. 6b). Namely, number fractions of highly hygroscopic mode particles were low in the early morning and evening ($nf_{highly} < 0.3$), and higher during afternoon (approaching the highest level of 0.6 around 15:00 LT). The value positively correlated with GF ($R = 0.97$). The number fraction of weakly hygroscopic mode particles was opposite of that for highly hygroscopic particles, and negatively correlated with mean $GF$ ($R = -0.95$). There was no clear diurnal variation for the number fraction of moderately hygroscopic mode particles (stable around 0.2). These results suggest that water uptake by wildfire haze particles are tightly related to the fractions of weakly and highly hygroscopic mode particles. The mean $g$ values for each mode were $1.05 \pm 0.02$ for weakly, $1.21 \pm 0.01$ for moderately, and $1.40 \pm 0.05$ for highly hygroscopic particles. The average values for $nf$ were $0.42 \pm 0.18$ for weakly, $0.18 \pm 0.07$ for moderately, and $0.40 \pm 0.20$ for highly hygroscopic particles (Table 4).

The diurnal variation in hygroscopic properties coincides with that in chemical composition (Fig. 6c). The mean $GF$ correlated well with $f_{SO4}{}^{2-}$, suggesting the primary role of $f_{SO4}{}^{2-}$ in governing the water uptake by wildfire haze particles. The enhancement of $f_{SO4}{}^{2-}$ accompanied decrease of $f_{org}$. Variation of chemical composition of organics also correlated well with water uptake properties. The fractional of signal intensity at $m/z$ 44 ($f_{44}$), which is considered as a marker ion for degree of oxidation (Duplissy et al., 2011; Ng et al., 2011; Ogawa et al., 2016), was also high during daytime, as in the case of mean $GF$ and $f_{SO4}{}^{2-}$. A similar pattern was also observed for $f_{OOA}$, while that for HOA was opposite. For instance, $f_{HOA}$ was the highest during morning rush hours, and subsequently decreased during daytime. Variation of the fresh PBOA fraction ($f_{PBOA}$) is kind of similar to that of non-peat BBOA ($f_{BBOA}$), i.e., without an apparent diurnal pattern during the severe wildfire haze periods.





1       The diurnal variation in change of organic composition was caused by enhanced $f_{OOA}$

during daytime, which accompanied decrease in $f_{HOA}$. In general, highly oxygenated organic
compounds are highly hygroscopic due to their water solubility, qualitatively explaining the
similarity in diurnal variations among mean $GF$, $f_{OOA}$, and $f_{44}$ (Duplissy et al., 2011; Zhao et al.,
2015; Ogawa et al., 2016). The relationship between particle hygroscopicity and degree of
oxidation of organics will be discussed in detail in Sect.5.2.

## 8  5. Discussions

### 9  5.1 Chemical composition dependences of water uptake by wildfire haze particles

Figure 7 depicts the relationships between $\kappa$ and mass fractions of both inorganic species
and organics in NR-PM$_1$ both for daytime and nighttime data. $\kappa$ and $f_{SO4^{2-}}$ positively correlate,
demonstrating that sulfate is the most important compound in governing water uptake by 100 nm
wildfire haze particles due to its high hygroscopicity. Similarly, $\kappa$ is positively related to $f_{NH4^{+}}$, as
it is the counter anion of sulfate. On the contrary, $\kappa$ negatively correlates with $f_{org}$, as organics are
less hygroscopic than inorganic salts. There is no clear correlation between $\kappa$ and $f_{NO3^{-}}$, implying
that nitrate is not the major contributor to the variability of $\kappa$. $\kappa$ is almost independent of $f_{Cl^{-}}$,
partially due to the limited availability of chloride.
The correlation between $\kappa$ and $f_{org}$ is relatively scattered. For instance, $\kappa$ can vary from
0.10 to 0.40 when $f_{org}$ is 0.7, meaning that some other factors than $f_{org}$ may also influence water
uptake. Variability in chemical characteristics of organics might have played a role in the scatter
of the data (Fig. 6d and e).

### 22  5.2 Relationship between hygroscopicity and chemical composition of organics

Table 2 summarizes the $\kappa_{org}$ and $\kappa_{OOA}$ results averaged over the same PM$_{2.5}$ filter
sampling periods. Figure 8 illustrates the relationship between $\kappa_{org}$ and $f_{WSOC}$. In general, $\kappa_{org}$
loosely correlates with $f_{WSOC}$ to some extent, except for the data on October 22[nd]. The relatively
high value of $f_{44}$ (0.11) on October 22[nd] might be the cause of the deviation. These results



indicate the importance of both the WSOC fraction and oxygenation degree of organics in the
hygroscopic growth.

3        Figure 9 depicts the relationships between $\kappa_{org}$ and $f_{44}$ as well as $\kappa_{org}$ and $f_{44}/f_{43}$. Daily

average data were utilized. Although the data are scattered, a positive correlation between $\kappa_{org}$
and $f_{44}$ was observed ($R = 0.70$). The signal of $m/z$ 44 (mostly $CO_2^+$) is known to originate from
highly oxidized organic functional groups such as dicarboxylic acids and acyl peroxides (Aiken
et al., 2008). These highly oxygenated functional groups contribute in promoting hygroscopicity
by enhancing water solubility ( Topping et al., 2005; Cubison et al., 2006; Hallquist et al., 2009;
Duplissy et al., 2011; Psichoudaki and Pandis, 2013; Suda et al., 2014; Riipinen et al., 2015;
Ogawa et al., 2016; Petters et al., 2016; Marsh et al., 2017). $\kappa_{org}$ and $f_{44}/f_{43}$ also presented a
similar trend to that of $\kappa_{org}$ and $f_{44}$. The correlations shown in Fig. 9 clearly demonstrate the
important role of oxygenation degree in water uptake properties of organic compounds in
wildfire haze particles.

14        During daytime, organic particles tend to be highly oxidized due to oxidation of primary

organic aerosol as well as formation of secondary organic aerosol from volatile organic
compounds (Fig. 6, Ng et al., 2010; Zhao et al., 2016). These chemical evolution processes of
organic aerosol particles will need to be better understood for quantitative prediction of water
uptake by wildfire haze particles (Kroll and Seinfeld, 2008; Riipinen et al., 2011; Winkler et al.,
2012; Ehn et al., 2014). The evolution process could induce alternation in size-dependence in
chemical composition as well as in mixing state (Chakrabarty et al., 2006; Zhao et al., 2015). To
unveil these unanswered questions, more details of the size- and mixing state dependent
chemical compositions as well as molecular level chemical characteristics of Indonesian wildfire
haze particles will be required.
**6. Conclusions**

26        In September − October 2015, Indonesian wildfire-induced transboundary haze pollution

spread through Southeast Asia, affecting both environment and climate dramatically and
ravaging public health and the economy seriously. As a downwind receptor city, Singapore




experienced the pervasive wildfire haze events. During the periods, we simultaneously measured
hygroscopic growth factors ($GF$) and chemical compositions of ambient wildfire haze particles
in Singapore, targeting for more comprehensive insights into the linkages between water uptake
and particle chemical composition as well as secondary aerosol formation.
High aerosol loading of non-refractory submicron particles (NR-PM$_1$, occasionally
exceeding 100.0 µg m$^{-3}$) was frequently observed, stressing the severity of the 2015 wildfire
haze pollution. The NR-PM$_1$ particles are predominantly composed of organics (OA,
approximately 77.1 % on average) and sulfate dominates the inorganic constituents (about
11.7 %). Chemical analyses of NR-PM$_1$ indicate the ubiquity and dominance of oxygenated
species in organics, in line with the most intense ion signals at $m/z$ 44 (mostly $CO_2^+$, $f_{44}$ = 10.5 %
in total OA mass) and $m/z$ 43 (most likely $C_2H_3O^+$, $f_{43}$ = 7.5 % on average). Moreover, a major
fraction of organics is water soluble, as signified by the high water-soluble organic carbon
fraction in ambient PM$_{2.5}$ filter samples (26.7 % of the total PM$_{2.5}$ mass).
Wildfire haze particles are generally highly hygroscopic. The hygroscopicity parameter, $\kappa$,
of 100 nm particles varies between 0.004 and 0.475, with a mean $\kappa$ value of 0.189 ± 0.087. The
derived mean $\kappa$ results of organics are 0.157 ± 0.108 ($\kappa_{org}$, bulk organics) and 0.287 ± 0.193
($\kappa_{OOA}$, oxygenated organic fraction), fairly comparable to the reported values for organic
compounds (Petters and Kreidenweis, 2007; Duplissy et al., 2011; Lathem et al., 2013; Cerully et
al., 2015). This highlights the difference in $\kappa$ between wildfire haze particles and fresh
Indonesian peat burning particles, which are intrinsically non-hygroscopic due to the rather
limited water-soluble organic fraction (Chen et al., 2017). $GF$ data show a notable diurnal
variation that commonly peaks during the daytime. This is identical with the diurnal pattern of
number fraction of highly hygroscopic mode particles and accompanied opposite fluctuation of
number fraction of weakly hygroscopic mode. These results imply the chemical composition
dependence of particle hygroscopicity, verified by the fact that $\kappa$ is positively correlated with
mass fraction of sulfate but inversely related to the mass fraction of organics. Aside from the
governing influence of sulfate, $\kappa$ of haze particles is also contributed by water uptake of organics.
$\kappa_{org}$ is loosely related to the water-soluble organic fraction, yet a positive correlation between $\kappa_{org}$
and $f_{44}$ was shown ($R$ = 0.70). This denotes that the oxygenation degree of organics may play an
important role in water uptake especially by organic-rich haze particles.



To our knowledge, this could be the first reported field water uptake measurements of
wildfire haze particles in Equatorial Asia. The presented results suggest that formation of
secondary aerosol particles, including both inorganics and organics, is the key in addressing the
variability in reported data on hygroscopic properties of aerosol particles originating from
Indonesian peatland fires. Size-dependent chemical composition as well as further detailed
chemical analysis will be needed in future studies for quantitative understanding on water uptake
by Indonesian wildfire-induced particles.
**Acknowledgements**
This work was supported by the Singapore National Research Foundation (NRF) under
its Singapore National Research Fellowship scheme (National Research Fellow Award,
NRF2012NRF-NRFF001-031), NRF Campus for Research Excellence and Technological
Enterprise (CREATE) program (NRF2016-ITCOO1-021), the Earth Observatory of Singapore,
and Nanyang Technological University. Takuma Miyakawa and Yuichi Komazaki were funded
by the Environment Research and Technology Development Fund (2-1403) of the Ministry of
Environment, Japan, and the Japan Society for the Promotion of Science (JSPS),
KAKENHI Grant number JP26550021. We would like to thank Wen-Chien Lee and Gissella B.
Lebron for assisting in the ambient haze observations. We also acknowledge the help of Pavel
Adamek in polishing our English.





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





**Table 1.** Summary of the hygroscopicity parameters ($\kappa$) and material densities of different
chemical constituents utilized in the theoretical $\kappa$ calculation with chemical data.

| Chemical compounds | Hygroscopicity parameter, $\kappa$ | Density (10³ kg/m³) |
|:---:|:---:|:---:|
| $SNA^{\dagger}$ | 0.59 | 1.77 |
| EC | 0 | 1.80 |
| OOA | $\kappa_{OOA}{}^{*}$ | 1.50 |
| Bulk OA | $\kappa_{org}{}^{*}$ | 1.40 |

$^{\dagger}$ *SNA* includes all the sulfate, nitrate, and ammonium in submicron wildfire haze particles.
* $\kappa_{OOA}$ and $\kappa_{org}$ were derived from ambient water uptake measurements and chemical data, in
combination of the given parameters in Table 1, using the simplified algorithm introduced in
Sects. 3.3 and 3.2.



1  **Table 2.** Summary of averaged chemical characteristics of the 24 h $PM_{2.5}$ filter samples collected

2  during 2015 haze events and the accordingly calculated mean $\kappa$ results of organics ($RH = 85$ %).

| Sampling date | $Cl^-$ ($\mu g/m^3$) | $NO_3^-$ ($\mu g/m^3$) | $SO_4^{2-}$ ($\mu g/m^3$) | $Na^+$ ($\mu g/m^3$) | $NH_4^+$ ($\mu g/m^3$) | $K^+$ ($\mu g/m^3$) | $Mg^{2+}$ ($\mu g/m^3$) | $Ca^{2+}$ ($\mu g/m^3$) |
|---|---|---|---|---|---|---|---|---|
| Oct-14 | 0.64 | 2.40 | 10.21 | 0.20 | 4.93 | 0.62 | 0.05 | 0.55 |
| Oct-15 | 0.38 | 1.72 | 10.51 | 0.18 | 4.63 | 0.60 | 0.03 | 0.50 |
| Oct-16 | 0.27 | 1.62 | 9.01 | 0.25 | 4.03 | 0.48 | 0.08 | 0.65 |
| Oct-19 | 0.65 | 2.39 | 15.90 | 0.20 | 7.99 | 0.58 | 0.03 | 0.41 |
| Oct-20 | 0.24 | 1.09 | 10.22 | 0.31 | 4.33 | 0.51 | 0.05 | 0.48 |
| Oct-21 | 0.19 | 0.83 | 7.95 | 0.23 | 3.15 | 0.51 | 0.08 | 0.52 |
| Oct-22 | 0.23 | 0.80 | 9.71 | 0.29 | 3.78 | 0.38 | 0.08 | 0.50 |
| Oct-23 | 0.56 | 2.65 | 15.92 | 0.18 | 9.13 | 0.70 | 0.005 | 0.25 |
| Average | 0.40 | 1.69 | 11.18 | 0.23 | 5.25 | 0.55 | 0.05 | 0.48 |

3  **Table 2. (continued)**

| Sampling date | $f_{inorg}$* (%) | $f_{SO4}^{2-}$ (%) | $f_{EC}$ (%) | WSOC/OC (%) | Mean $\kappa$ | Mean $\kappa_{org}$† ($D_0 = 100$ nm) | Mean $\kappa_{OOA}$† |
|---|---|---|---|---|---|---|---|
| Oct-14 | 42.1 | 21.9 | 15.3 | 79.9 | 0.114 | 0.030 | 0.054 |
| Oct-15 | 41.1 | 23.3 | 11.1 | 64.8 | 0.202 | 0.118 | 0.185 |
| Oct-16 | 28.6 | 15.7 | 11.7 | 66.4 | 0.161 | 0.133 | 0.256 |
| Oct-19 | 26.6 | 15.0 | 5.9 | 60.2 | 0.265 | 0.229 | 0.450 |
| Oct-20 | 30.3 | 18.0 | 11.3 | 61.5 | 0.271 | 0.236 | 0.443 |
| Oct-21 | 26.4 | 15.6 | 11.0 | 56.9 | 0.193 | 0.148 | 0.283 |





| | | | | | | | |
|---|---|---|---|---|---|---|---|
| **Oct-22** | 42.9 | 26.4 | 15.8 | 59.5 | 0.265 | 0.212 | 0.376 |
| **Oct-23** | 24.6 | 13.3 | 4.4 | 59.6 | 0.217 | 0.184 | 0.337 |
| **Average** | 30.5 | 17.2 | 10.8 | 63.6 | 0.211 | 0.161 | 0.298 |

\* The subscript $_{inorg}$ stands for all the inorganic species; hence $f_{inorg}$ is the mass fraction of
inorganic particles in the $PM_{2.5}$ filter sample. All the ionic data are provided with the mean mass
concentration ($\mu g/m^3$).
$^{\dagger}$ The mean $\kappa$ results of organics here were calculated with assuming 10.0 % elemental carbon
(*EC*) in total mass (see Sect.3.2).



1  **Table 3.** Derived mean $\kappa$ values of organics with consideration of 10.0 % *EC* mass fraction in

2  total NR-PM$_1$, as well as the mean $\kappa$ results for HTDMA measurements conducted within the

3  overlapped ToF-ACSM observation period of $10 - 24$ October 2015 (i.e., $\kappa_{HTDMA}$).

| | Mean $\kappa$ (mean ± SD) | | |
|---|---|---|---|
| | **Overall** | **Daytime** | **Nighttime** |
| $\kappa_{org}$ | $0.157 \pm 0.108$ | $0.200 \pm 0.104$ | $0.103 \pm 0.086$ |
| $\kappa_{OOA}$ | $0.287 \pm 0.193$ | $0.353 \pm 0.179$ | $0.206 \pm 0.178$ |
| $\kappa_{HTDMA}$ | $0.207 \pm 0.093$ | $0.247 \pm 0.096$ | $0.160 \pm 0.063$ |





**Table 4.** The temporally mean number fraction ($nf$), volume-weighted mean diameter growth
factor ($GF$), and $\kappa$ results (mean ± SD) of 100 nm wildfire haze particles in the three different
hygroscopicity ranges at 85 % $RH$.

| Hygroscopic mode | $nf$ | $GF$ | $\kappa$ |
|---|---|---|---|
| **Weakly** ($g < 1.15$) | $0.42 \pm 0.18$ | $1.05 \pm 0.02$ | $0.030 \pm 0.013$ |
| **Moderately** ($1.15 \le g < 1.27$) | $0.18 \pm 0.07$ | $1.21 \pm 0.01$ | $0.151 \pm 0.005$ |
| **Highly** ($1.27 \le g < 1.85$) | $0.40 \pm 0.20$ | $1.40 \pm 0.05$ | $0.343 \pm 0.054$ |
| **Bulk mean** | $n/a$ | $1.25 \pm 0.09$ | $0.189 \pm 0.087$ |

$n/a$: not applicable.



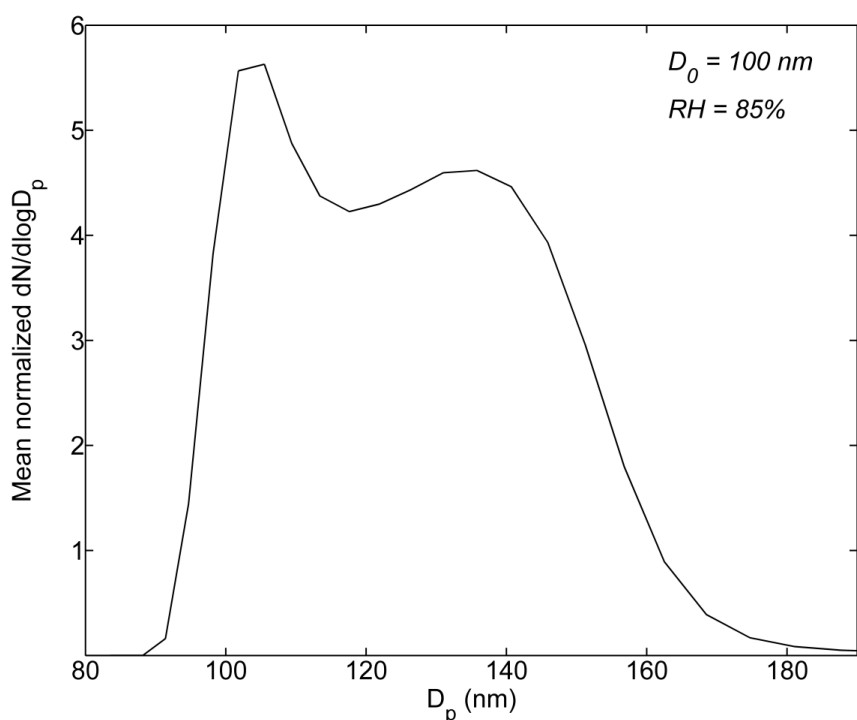

2 **Fig.1** Normalized particle number size distribution ((dN/dlogDp)/N) after humidification

3 averaged over the whole haze observation period.



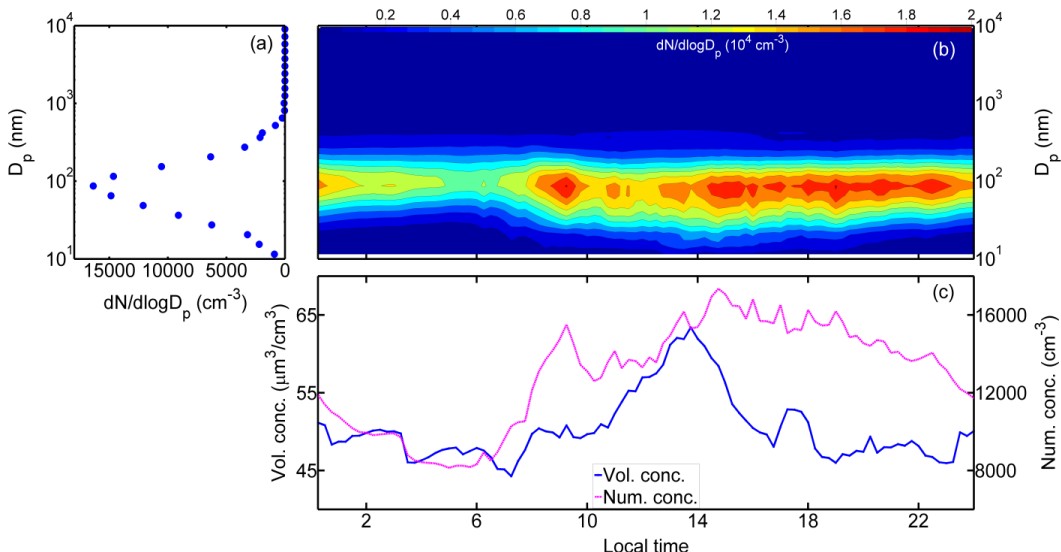

**Fig.2** (a) The mean normalized particle number size distribution ($dN/dlogD_p$, $cm^{-3}$) during the

ambient wildfire haze observations, as well as the mean diurnal variations of (b) normalized size

distribution, (c) number concentration (Num. conc., $cm^{-3}$, denoted by the magenta solid line) and

volume concentration (Vol. conc., $\mu m^3/cm^3$, as the blue line displayed) measured with

NanoScan-SMPS ($11.5 - 365.2$ nm) and OPS ($419$ nm $- 10$ $\mu$m) (local time, LT).

High aerosol loading was commonly observed during the transboundary wildfire haze.

Submicron particles in the size range of $30 - 200$ nm dominated the total number concentration

of wildfire haze particles.





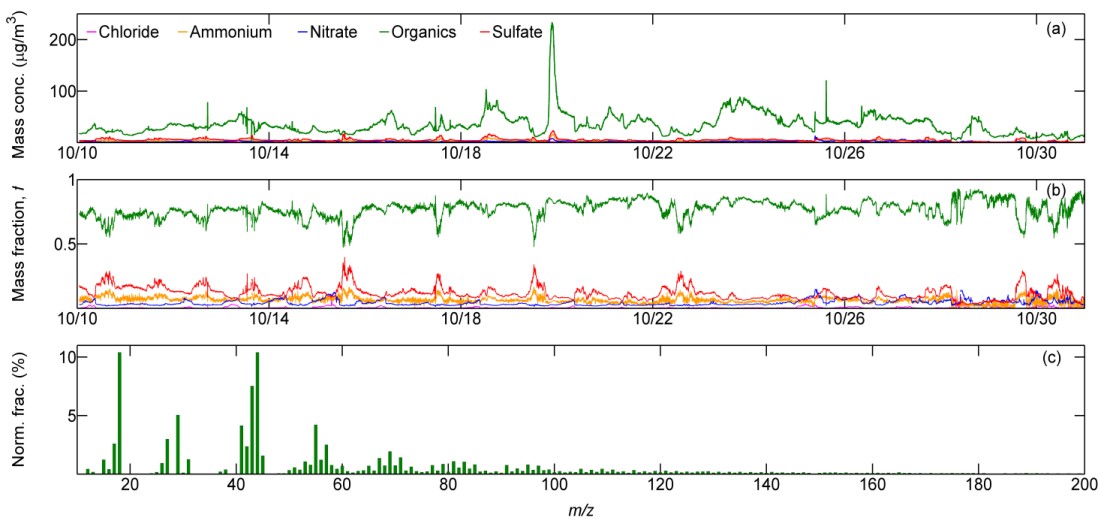

**Fig.3** Time series of (a) mass concentration (Mass conc., $\mu g/m^3$) and (b) corresponding mass
fraction, $f$, of the five specific chemical species in NR-PM$_1$ measured by ToF-ACSM (abscissa
shows the observation dates in October 2015, with the date format of Month/Day). (c)
Temporally averaged OA mass spectra for submicron wildfire haze particles, displayed with the
normalized ion fraction (Norm. frac., %) of each ion fragment.
Wildfire haze particles were predominantly composed of organics. Ion signals ($m/z$) from
oxygenated organics (e.g., $m/z$ 43, 44) were prominent, while intensities of ions for hydrocarbon-
like organic compounds (e.g., $m/z$ 41, 55, 57) and biomass burning tracers (e.g., $m/z$ 60, 73) were
relatively less intense. See the text for details.

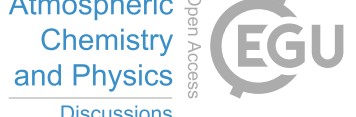



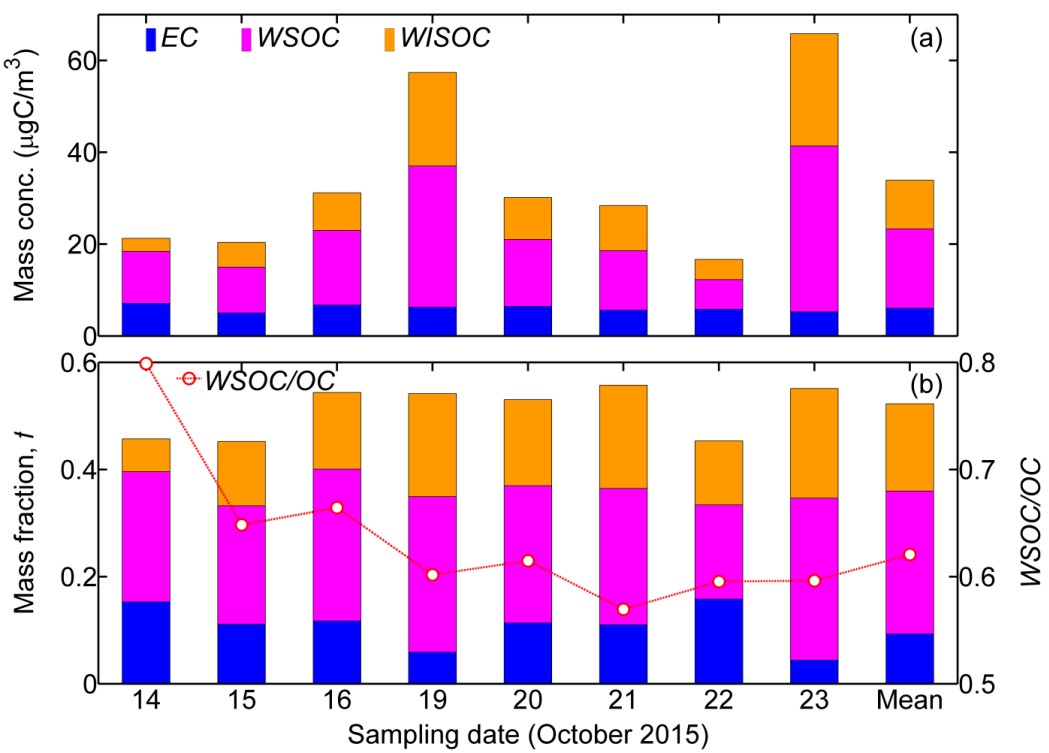

**Fig.4** (a) Mass concentration (Mass conc., μgC/m$^3$) and (b) corresponding mass fraction, $f$, of the
carbon contents including *EC*, *WSOC*, and water-insoluble *OC* (*WISOC*) in *PM$_{2.5}$* filter samples.
The *WSOC/OC* ratio is also displayed by the scattered dots in panel (b). All the corresponding
temporal mean results are shown as denoted by "Mean".
The *WSOC* fraction was exclusively higher than that of *WISOC* or *EC*, highlighting the
dominance of *WSOC* in the carbon content of *PM$_{2.5}$* filter samples. In general, the *EC* fraction
fluctuates around 10.0 % of the total *PM$_{2.5}$* mass.



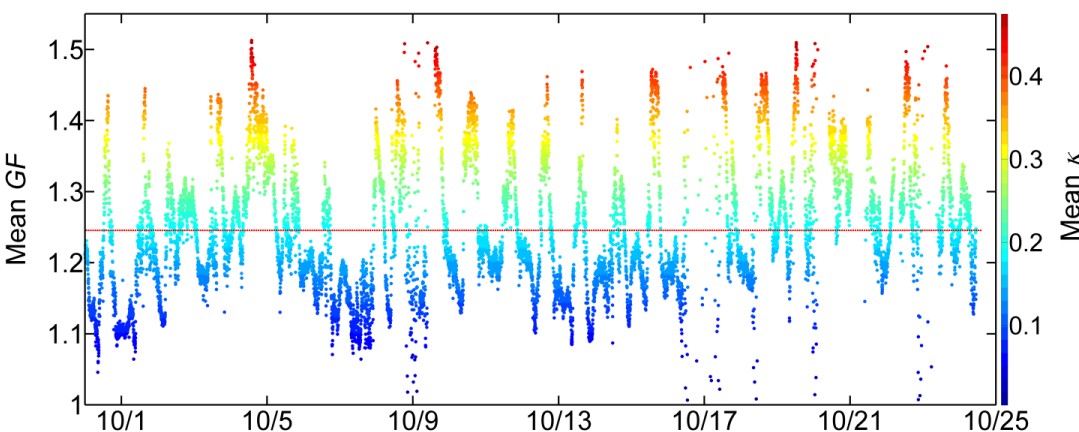

**Fig.5** Time series of the volume-weighted mean particle diameter growth factor (*GF*) derived
from HTDMA measurements (date format: Month/Day, 2015), colored with the corresponding
mean hygroscopicity parameter, $\kappa$. The red dashed line stands for the temporal mean *GF*
averaged over the whole observation period.



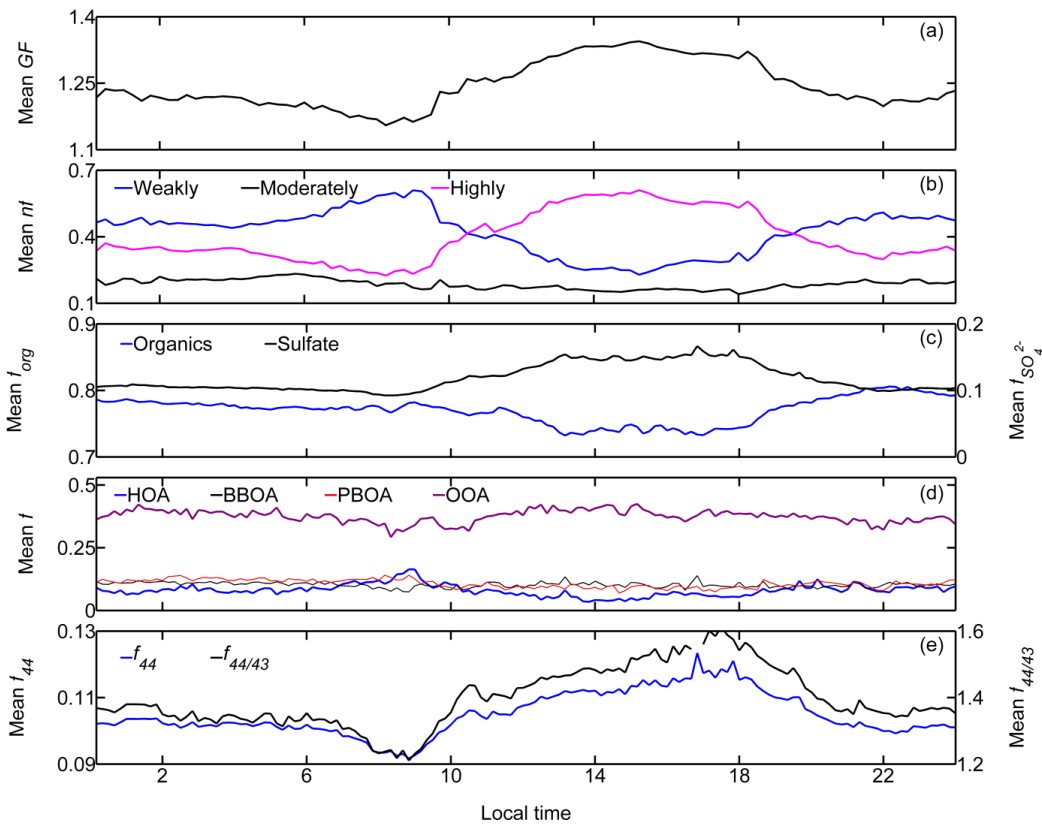

**Fig.6** Diurnal variations of (a) the mean *GF*, (b) number fraction (*nf*) of the three hygroscopic
modes, (c) mass fraction of the two main components in NR-PM$_1$, i.e., organics ($f_{org}$) and sulfate
($f_{SO4}^{2-}$), (d) mass fraction of ME2-resolved four OA factors in NR-PM$_1$ (i.e., $f_{HOA}$, $f_{BBOA}$, $f_{PBOA}$,
and $f_{OOA}$), and (e) mean $f_{44}$ and $f_{44/43}$ of organics in NR-PM$_1$ (local time, LT).

Pronounced diurnal patterns were observed for the mean *GF*, number fractions of both weakly
and highly hygroscopic modes, $f_{44}$, $f_{44/43}$, and mass fractions of organics and sulfate as well as
HOA and OOA. $nf_{highly}$, $f_{SO4}^{2-}$, $f_{44}$, $f_{44/43}$, and $f_{OOA}$ show the similar variations to that of the mean
*GF*, whereas contrary diurnal patterns are found for $nf_{weakly}$, $f_{org}$, and $f_{HOA}$.





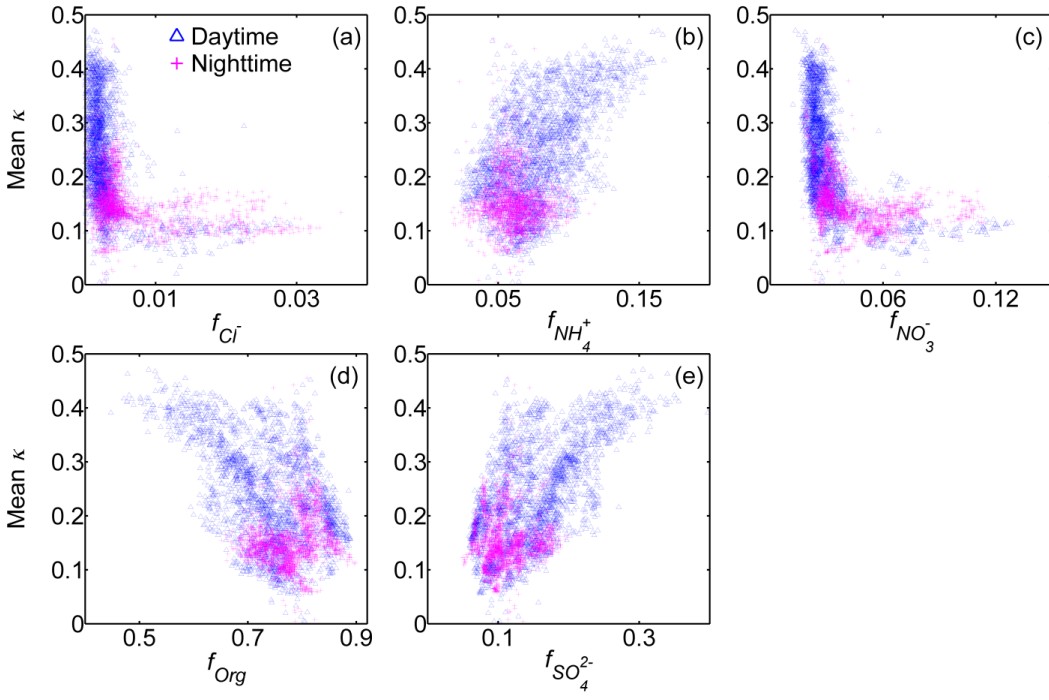

2   **Fig.7** Relationships between the mean $\kappa$ results (100 nm) and mass fractions of the five non-

3   refractory chemical compositions in submicron wildfire haze particles.





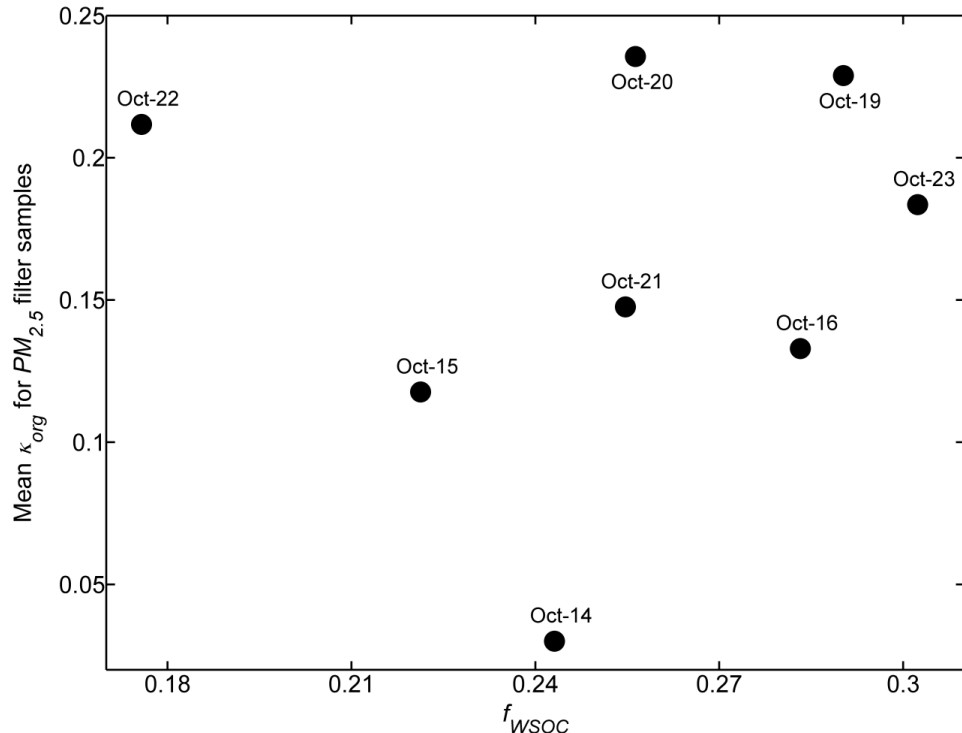

2  **Fig.8** Correlation between the mean $\kappa$ of organics ($\kappa_{org}$, with 10.0 % *EC* mass fraction taken into

3  consideration of the $\kappa$ calculation) and the mean *WSOC* fraction ($f_{WSOC}$) of *PM$_{2.5}$* filter samples.



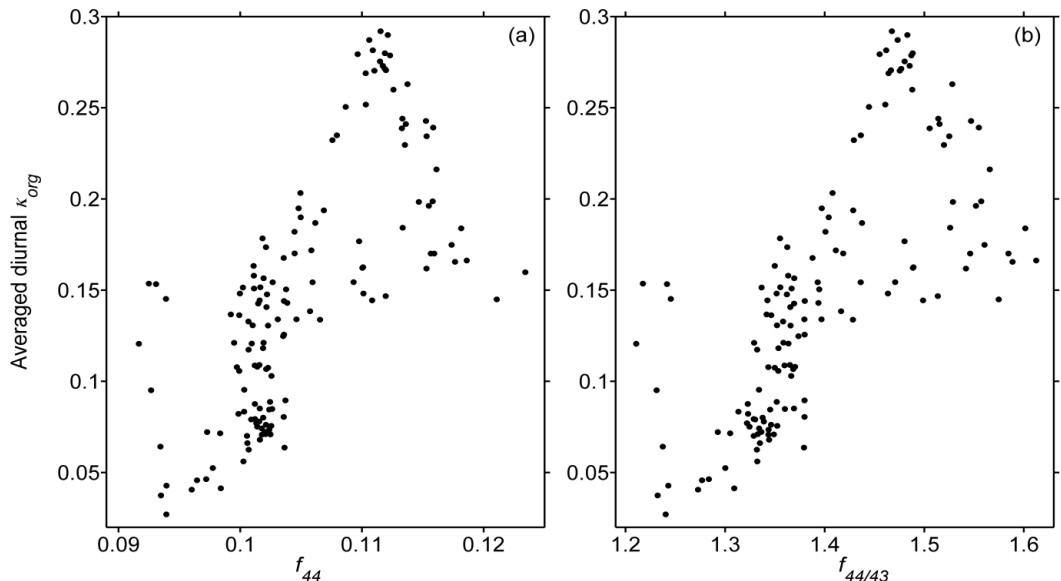

**Fig.9** Relationships between the averaged diurnal $\kappa_{org}$ results vs. (a) $f_{44}$ and (b) $f_{44/43}$ in NR-PM$_1$

3    haze particles.