# Peer review of "Secondary aerosol formation promotes water uptake"

_Atmospheric Chemistry and Physics, 2017_

## Referee Comment (RC1) · Anonymous Referee #1 · 11 Nov 2017

Review of Chen et al. (2017) ACP

Summary

The paper presents measurements of biomass burning aerosol hygroscopicity, both fresh and with aging. This is an important region and biome to characterize for smoke properties. The paper finds a substantial difference in kappa attributed to difference in the organic fraction hygroscopicity. The latter is related to aging and the fraction of water-soluble organic material. The paper is a well referenced manuscript and the methods appear sound. A few items need further attention before this publishable in ACP.

[Figure]

Major Comments on Content

Line 6 is it agricultural development in this region? Housing, industrial? More specifics are recommended.

Show some data on calibration in the paper or supplementary material. This is important to give the reader confidence in the findings. The reference to the first paper is a help but some indication of these efforts is merited in the paper or supplementary material.

I may have missed it, but do you apply any OC to particulate organic carbon multiplier or use as measured OC? This would be good to state up front and/or in abstract.

Relying on the publication of a manuscript in preparation for experimental details is somewhat risky (Budisulistorini et al.).

Do you have any indication of age of the smoke aerosols that were sampled? Any characterization of the combustion characteristics of the surrounding fires (fuels, phase, etc.)? Presumably a smoldering dominated combustion situation with peat burning.

Jayaranthe et al. (2017) on the organic carbon content and WSOC of SE Asian peatland smoke is a relevant reference to bring into your analysis.

The strength of the hygroscopic response of peat smoke organic aerosol as it ages is somewhat surprising. The assertion that wildfire haze particles are generally highly hygroscopic in the conclusion counters most of the research in the literature. Compare to the results from the Carnegie-Mellon group (e.g. Englart et al. 2012) or FLAME results (e.g. Carrico et al. 2010). Relevant fuels such as duff show little water uptake and only modest increases with oxidation (kappa < 0.1) for organic carbon dominated aerosol from this sourced. Commenting on this is useful.

How is RH obtained with possible temperature changes in the system? At the high RH $\sim$85% of your measurements a 1degC change in temperature results in 5% change in RH. Accessing the cited Chen et al. (2017) the RH was monitored at the inlet and

outlet of the humified DMA. Could the temperature of the column have been different?

Related, could the diurnal variation be due to changes in sampling conditions? I find it surprising that the organic fraction hygroscopicity would change so much day to night (table 3, fig 5). More discussion as to what you attribute the changes would improve. If photochemistry and oxidation of organics is suggested is there any relationship to solar input on the days of sampling?

To what do you attribute the decrease in kappa with increasing chloride and nitrate fraction? Shouldn't these be hygroscopic inorganic species that would contribute to increasing kappa? Why would kappa increase so strongly as these ion fractions approach zero?

A relationship in figure 9 is likely but figure 8 shows little relationship (suggest dropping this and stating no relationship found). Even without Oct 22 the relationship looks weak, r2 of 0.2 or so?

Comments on Presentation

Overall the writing mechanics is reasonable and clear but needs further work for ACP standards. I will highlight a few passages with suggestions for better writing. I recommend further refinement with a fluent English writer to raise it to acceptable levels. Here are a few:

Line 14 change to "not only in terms of hazards to human health"

Line 16 I'm not aware of any prizes offered for poor air quality and though I'm sure the region would vie for such a title I recommend keeping magazine-like statements out of the article.

In Figure 5 I suggest either plotting d/do or kappa an not the rainbow colors of kappa that follows the same pattern as d/do that is plotted.

The traces in figure 6 are somewhat difficult to distinguish.

Chemical characterization of fine particulate matter emitted by peat fires in Central Kalimantan, Indonesia, during the 2015 El Niño, Jayarathne et al. (2017) ACP.

Cloud condensation nuclei activity of fresh primary and aged biomass burning aerosol, G. J. Engelhart, Atmos. Chem. Phys., 12, 7285–7293, 2012, www.atmos-chem-phys.net/12/7285/2012/doi:10.5194/acp-12-7285-2012

---

## Referee Comment (RC2) · Anonymous Referee #2 · 12 Jan 2018

This manuscript describes an aerosol dataset collected in Singapore during a period in October, 2015 when the region was impacted by high concentrations of smoke particles. Though I have some concerns with the analysis, I support eventual publication in ACP in part because the region and this particle source are understudied. But major revisions would be required first. In addition to the concerns identified below, the writing would need to be improved prior to publication. The mistakes are too numerous to identify in this review.

A general concern I have with the manuscript is that it lacks a description of the meteorological and chemical setting that readers would need to understand the data. There

is a brief mention of the location of the sampling site relative to a road and a petro-chemical complex, but no discussion about whether and when those sources would be upwind. There is also no discussion of the typical transport time of the smoke prior to arrival at the sampling site. And to put the smoke-impacted measurements into context, there needs to be some data and discussion of the typical concentration and composition at the site.

Line 20 (and later): There is no discussion about the disconnect between the single size at which g was measured (100 nm) and the larger particles that dominate the composition measured by the ACSM. There is a mention that the number size distribution is dominated by particles in the 50 – 200 nm size range, but the number distribution is irrelevant. It seems likely that the diurnal variations in g would not be as important for particles near the mass median size of the distribution. So then the composition should be comparatively constant and the variation between it and g introduces scatter to the inferred hygroscopicity parameters. It also seems likely that much of the sulfate will be in >100 nm particles if it forms in cloud. So then it is questionable how much direct impact it has on g.

I am unfamiliar with the characteristics of PBOA. Is it reliably separated from BBOA? Could variation in fuel type or burning characteristics cause shifts in attribution between the two types? And what is known about the solubility of peat burning primary particles? I ask because it could influence the fraction of soluble species in solution at the 85% RH in the HTDMA. These points should be added to the manuscript and not simply provided in a response.

Page 8, line 13: The OPS does not measure aerodynamic size.

Page 9, Line 3: The only g distribution provided is the study average shown in Figure 1. The separation of particles into the three g categories implies that the distributions were generally multimodal. I don't doubt that, but it should be shown. Perhaps a few example distributions could be included in a supplemental document.

[Figure]

Page 9, line 14: I think "More" would be a better word here. Highly to me implies much higher g than 1.27. Especially when those with g < 1.15 are "Nearly non-hygroscopic"; that's a large change in type over a small change in g.

Page 10, line 9: Those are not salts

Page 10, line 12: Is sea salt expected to be important at the site? Unless operated at a high temperature the ACSM would not see it.

Page 10, line 21: The second sentence in this paragraph needs to be rewritten.

Page 11, line 15: I can't understand why the authors chose to assume that both BBOA and PBOA have kappas of 0.0. Even if they are low, why not use best estimates of the values instead of just arbitrarily setting them to 0. Because they are not actually completely non-hygroscopic, the result will be an erroneous sensitivity of kappa_OOA to the organic type fractions. This needs to be changed in the revision.

Page 12, line21: I agree that most of wildfire haze mass will be submicron, but that can't be asserted based on the number concentration as the authors do. At a minimum the supermicron volume fraction should be used for this conclusion.

Page 12, lines 25 and 26: dN/dlogDp is not the same as number concentration.

Page 13, line 1: The authors note that traffic emissions may be important but take no steps to correct for the impact in their analysis. Should periods when winds are from the road and when traffic is heavy be excluded? The novelty of this dataset is obviously the wildfire haze and anything local is an interference.

Page 13, line 22: Add non-refractory in front of each instance of chloride.

Page 14, line 2: Related to the above comment about the assumption that only the OOA contributed to hygroscopicity, the fraction of the organic mass that was water soluble was quite high (average 64%, max 80%). How frequently was this higher than the fraction of organics categorized as OOA?

Page 14, line 7: Is it surprising that there is much less K than SO4?

Page 14, line 26: Somewhere in the paper there needs to be a conceptual explanation of what is responsible for the observed mix of HOA, PBOA, BBOA, and OOA. Do the authors believe that closer to the source measurements would show mostly PBOA and BBOA and that aging is responsible for the conversion to OOA? (the transport timescale issue mentioned above is relevant here). Is the daily variation thought to be mostly due to photochemical processing or could it also be daily variation in fire characteristics? And is it thought that the changes are due to processing of the existing particles or production of SOA from gas phase emissions from the fires? The one example size distribution time series presented does not seem to indicate particles are growing, but rather the size is about constant and only the concentration changes. Do the time series on other days look like that as well? And if so, what is responsible?

Page 17, line 15: NO3 was a small contributor to the overall mass. It seems pointless to note that variations in that small contribution had little effect on kappa.

Figure 3: I would like to be able to see whether organic mass and sulfate mass are correlated. But with the use of the same y-scale for both in (a) that is not possible.

Figure 6: Specify date

Figure 9: Clarify what "averaged diurnal" means.
* * *

---

## Author Comment (AC1) · 23 Feb 2018

Dear Editor,

We would like to thank the two reviewers for their constructive comments and suggestions, which have been fully taken into account upon manuscript revision. A point-by-point response and an accordingly updated manuscript have been uploaded.

In the following, original reviewer comments, our response, and updates on the revised manuscript are shown in **bold**, normal, and *italic*, respectively.

Kind Regards,

Jing Chen, Mikinori Kuwata

**Anonymous Referee #1**

**General comments:**

**R1C0: The paper presents measurements of biomass burning aerosol hygroscopicity, both fresh and with aging. This is an important region and biome to characterize for smoke properties. The paper finds a substantial difference in kappa attributed to difference in the organic fraction hygroscopicity. The latter is related to aging and the fraction of water-soluble organic material. The paper is a well referenced manuscript and the methods appear sound. A few items need further attention before this publishable in ACP.**

**Response:** We appreciate the reviewer for insightful comments in revising the manuscript. Our responses to reviewer's concerns are described in the following.

**Major Comments on Content**

**R1C1: Line 6 is it agricultural development in this region? Housing, industrial? More specifics are recommended.**

**Response:** We appreciate for the helpful comment. Over the past few decades, Indonesia has experienced rapid land use change due to development of plantations as well as agricultural activities of small farmers (Miettinen et al., 2012; Marlier et al., 2015). Especially, development of peatland has been playing an important role for wildfire, as peat becomes highly flammable after drainage of water for agricultural development. The following sentences were added to the revised manuscript for clarification.

***Page 3, Line 5:*** *'The increasing wildfire activity is associated with the recent rapid change in land use for agricultural development, including industrial plantations over peatland (Page et al., 2009; Marlier et al., 2015; Spracklen et al., 2015). Such developments are accompanied by the drainage of water of pristine peat swamp forest, making the tropical peatland susceptible to fire (Langner et al., 2007; Konecny et al., 2016).'*

**R1C2: Show some data on calibration in the paper or supplementary material. This is important to give the reader confidence in the findings. The reference to the first paper is a help but some indication of these efforts is merited in the paper or supplementary material.**

**Response:** Calibration data of HTDMA by ammonium sulfate particles were added to supplemental material of the revised manuscript to address the reviewers' concern. Our calibration result agreed well with a literature data (Tang and Munkelwitz, 1994).

[Figure]

*Fig. S1 Comparison of particle diameter growth factor (GF) results derived from HTDMA calibration data for 150 nm dry ammonium sulfate (AS) particles, and experimental data (without parameterization) obtained by Tang and Munkelwitz (1994).*

*Page 7, Line 9: '... T is the absolute temperature (298 K). The HTDMA calibration results with 150 nm ammonium sulfate particles are shown in Fig. S2, demonstrating the validity of our instrument. Further details about the HTDMA are available in Chen et al. (2017).'*

**R1C3: I may have missed it, but do you apply any OC to particulate organic carbon multiplier or use as measured OC? This would be good to state up front and/or in abstract.**

**Response:** Yes, we employed an organic matter to OC ratio of 1.4 in calculating mass concentrations of total organics from OC content of the $PM_{2.5}$ filter samples, as detailed in section 4.2 (see Page 14, line 29). The derived organics mass concentration of $PM_{2.5}$ filter samples correlated well with organics quantified by the ToF-ACSM.

**R1C4: Relying on the publication of a manuscript in preparation for experimental details is somewhat risky (Budisulistiorini et al.).**

**Response:** We appreciate the reviewer for the comment. We recently submitted the corresponding manuscript, and have updated the citations in the present manuscript.

*Page 7, Line 23: '... Details about the ToF-ACSM measurements and data analysis are provided in Budisulistiorini et al. (submitted to Atmospheric Chemistry and Physics, February 2018).'*

*Page 15, Line 2: '... and the lack of EC content for the ToF-ACSM measurements (Budisulistiorini et al., submitted to Atmospheric Chemistry and Physics, February 2018).'*

*Budisulistiorini, S. H., Riva, M., Williams, M., Miyakawa, T., Chen, J., Itoh, M., Surratt, J. D., and Kuwata, M.: Dominant contribution of oxygenated organic aerosol to haze particles from real-time observation at Singapore during an Indonesian wildfire event in 2015. Submitted to Atmospheric Chemistry and Physics, February 2018.*

**R1C5: Do you have any indication of age of the smoke aerosols that were sampled?**

**Response:** We thank the reviewer for the helpful comments. In the revised manuscript, we added distributions of carbon emissions from wildfire (GFEDv4) as well as back trajectories of air masses arriving at Singapore. It took approximately 1~2 days for air mass transporting from the Southern part of Sumatra Island to Singapore, and 3~4 days for wildfire plumes from Central Kalimantan to arrive at Singapore.

[Figure]

***Fig. S1*** *Back trajectories of air masses arriving at Singapore and monthly carbon emission from wildfires during the observation period of HTDMA measurements. The transport time of wildfire plumes is approximately 1−2 days from South Sumatra, and 3−4 days from Central Kalimantan arriving at Singapore.*

*The back trajectory was calculated using the NOAA HYSPLIT model at 500 m (Kalnay et al., 1996). The altitude of the trajectories was constrained as iso-sigma. Carbon emission data was from the Global Fire Emissions Database (GFEDv4, https://daac.ornl.gov/VEGETATION/guides/fire_emissions_v4.html).*

**R1C6. Any characterization of the combustion characteristics of the surrounding fires (fuels, phase, etc.)? Presumably a smoldering dominated combustion situation with peat burning.**

**Response:** Based on previous laboratory studies and field observations, Indonesian peatland fires are predominantly contributed by smoldering combustion of underground organic-rich peat soils. Vegetation burning, which could include both

flaming and smoldering, also contributes to the emissions of gas and particulate matters (Field et al., 2016; Page et al., 2009; Jayaranthe et al., 2017).

Following the suggestion, these points are clarified into Sect. 2.1 of the revised manuscript, as detailed in the following.

*Page 6, Line 7:* '*... located approximately 8 km south. In September−October 2015, the observation site encountered severe transboundary haze pollution that was caused by recurring Indonesian peatland fires, dominated by the smoldering combustion of underground organic-rich peat soils and mixed surface vegetation burning (Field et al., 2016; Page et al., 2009; Jayaranthe et al., 2017). Particles emitted from the wildfires had experienced for approximately 1−4 days atmospheric aging process before arriving at Singapore (Fig. S1).*'

**R1C7: Jayaranthe et al. (2017) on the organic carbon content and WSOC of SE Asian peatland smoke is a relevant reference to bring into your analysis.**

**Response:** We thank the reviewer for the suggestion. Jayaranthe et al. (2017) quantified the mean WSOC/OC ratio of fresh peatland burning particles at Kalimantan in 2015 as 16 %. The value of our data in Singapore (63.6 %) is significantly higher than that reported in Jayaranthe et al. (2017), suggesting that secondary formation as well as chemical transformation of organic aerosol during atmospheric transport from Kalimantan to Singapore is important. We have added the following content into the discussion to emphasize the point.

*Page 14, Line 16:* '*... and maximum values of 63.6 % and 79.9 %, respectively. The mean WSOC/OC result was significantly higher than that for fresh Indonesian peat burning particles emitted from the source region (i.e., 16 %; Jayaranthe et al.,*

*2017), which were demonstrated to be generally water insoluble and thus nearly non-hygroscopic (Chen et al., 2017). This result suggests that the majority of organics in the wildfire haze particles were water soluble, implying the importance of secondary formation as well as the chemical transformation of organic particles during atmospheric transport.'*

**R1C8: The strength of the hygroscopic response of peat smoke organic aerosol as it ages is somewhat surprising. The assertion that wildfire haze particles are generally highly hygroscopic in the conclusion counters most of the research in the literature. Compare to the results from the Carnegie-Mellon group (e.g. Engelhart et al. 2012) or FLAME results (e.g. Carrico et al. 2010). Relevant fuels such as duff show little water uptake and only modest increases with oxidation (kappa < 0.1) for organic carbon dominated aerosol from this sourced. Commenting on this is useful.**

**Response:** We appreciate the reviewer for the critical comment. One important difference between our observation and former laboratory studies is the time scale of chemical processing. As discussed in R1C5, it likely took 1 ~ 4 days for particles originating from peatland fires to be transported to Singapore. On the other side, laboratory experiments are typically conducted for a time scale of hours (Chang et al., 2010; Engelhart et al., 2012). Laboratory chamber experiments are conducted at a relatively lower humidity (RH < 70%) in most cases, while Southeast Asia is always highly humid. As high RH is favorable for SOA formation by particle phase chemical reactions as well as for POA aging (Lambe et al., 2011a, b, 2015), the difference in RH might be playing an important role in the difference between observation and experimental data. In addition, concentrations

of other gas species such as $NO_x$ and sulfate, those are known to influence gas and particle phase chemistry (Kang et al., 2007; Chan et al., 2009; Lambe et al., 2011a, 2015), might be different between previous laboratory experiments and field condition. Furthermore, the difference in VOCs emitted from peat burning could be different from those emitted from other types of fires (Stockwell et al., 2016).

Nevertheless, our temporal mean $\kappa$ value of the OOA factor (i.e., $0.266 \pm 0.184$) for wildfire haze particles is still comparable to that of aged biomass burning particles (i.e., $0.2 \pm 0.1$, Engelhart et al., 2012). Following the reviewers' suggestion, the following description was added to the revised manuscript.

***Page 15, Line 27:*** *'...The derived $\kappa_{org}$ results are demonstrated to be comparable to previously reported $\kappa$ values for bulk organics (Petters and Kreidenweis, 2007; Duplissy et al., 2011; Lathem et al., 2013; Cerully et al., 2015). Moreover, the mean $\kappa_{OOA}$ value agreed well with mean $\kappa$ results in previous field and laboratory studies (e.g., $\kappa = 0.25 \pm 0.06$ for biogenic OOA, Chang et al., 2010; $\kappa = 0.2 \pm 0.1$ for aged BBOA, Engelhart et al., 2012). A caveat of this comparison is the representativeness of laboratory experiments for the actual environmental conditions, including types of burnt biomasses, concentrations and types of oxidants, and aging time. As the atmospheric condition of tropical Asia is unique, a systematic study of the chemical aging process of wildfire plume for the region would be required in the future. Note that the mean $\kappa_{OOA}$ was even higher than the corresponding mean $\kappa_{HTDMA}$ ($0.207 \pm 0.093$), revealing that the water uptake particularly by some highly oxygenated organics in the wildfire haze particles could be highly significant.'*

**R1C9: How is RH obtained with possible temperature changes in the system? At the high RH ~ 85% of your measurements a 1degC change in temperature results in 5% change in RH. Accessing the cited Chen et al. (2017) the RH was monitored at the inlet and outlet of the humidified DMA. Could the temperature of the column have been different?**

**Response:** In our study, RH of the humidified sheath outflow was modulated to be the target RH via PID control. For the whole humidification unit, the humidified DMA column and tubing connecting with the Nafion humidifier were wrapped well with thermal-insulation materials. There is a small temperature variance in the outer wall of the humidified DMA column (~ 0.3 °C lower in maximum) and the humidified sheath flow. The caused RH fluctuation is insignificant to the resultant GF variation, which is supported by our reasonable HTDMA calibration results with 150 nm ammonium sulfate particles (also see our response to **R1C2**).

**R1C10: Related, could the diurnal variation be due to changes in sampling conditions? I find it surprising that the organic fraction hygroscopicity would change so much day to night (table 3, fig 5). More discussion as to what you attribute the changes would improve. If photochemistry and oxidation of organics is suggested is there any relationship to solar input on the days of sampling?**

**Response:** We appreciate the reviewer for the insightful comment. The temperature of our laboratory was well-maintained by an air-conditioning system. There was no significant difference in RH of the HTDMA during daytime (85% ± 0.2 ~ 0.3%) and nighttime (85% ± 0.2 ~ 0.4%), suggesting that the diurnal variation is not caused by fluctuation in the instrumental condition.

The data in Fig. 5 is for mean GF, meaning that it includes contributions of both organics and inorganics to the bulk hygroscopic growth. As shown in Figs. 6 and 7, one of the key drivers for the diurnal variation is the enhanced sulfate fraction during the daytime (i.e., change in hygroscopicity of organics is not the only one reason for the diurnal variation of mean GF). Of course, enhanced fraction of oxygenated organics during daytime also contributed to the diurnal variation in mean GF.

The corresponding description in the manuscript was updated as follows to clarify these points.

***Page 17, Line 20:*** *'…Variations in the fresh PBOA fraction ($f_{PBOA}$) were similar to those of non-peat BBOA ($f_{BBOA}$) (Fig. 6e), namely, there was no apparent diurnal pattern during the severe wildfire haze periods. Consequently, the distinctly enhanced mean GF during the day could be attributed to the increase in both $f_{SO4}{}^{2-}$ and $f_{OOA.}$'*

**R1C11: To what do you attribute the decrease in kappa with increasing chloride and nitrate fraction? Shouldn't these be hygroscopic inorganic species that would contribute to increasing kappa? Why would kappa increase so strongly as these ion fractions approach zero?**

**Response:** We appreciate the reviewer for pointing it out. As we can see in Fig.7, both chloride and nitrate are enriched during nighttime, while availability of these species is limited during daytime. As both ammonium nitrate and ammonium chloride tend to partition to particle phase at a high RH environment, contributions of these species are pronounced during nighttime (Aan de Brugh et al., 2012; Gong

et al., 2013). On the other hand, their abundances in particle phase are limited during daytime because the corresponding RH is lower than that for nighttime. Particles tend to be more hygroscopic during daytime due to in-situ formation of sulfate. In summary, sulfate formation during daytime and diurnal variation in gas-particle partitioning of nitrate and chloride are producing the apparent anti-correlations. This point is clarified in the revised manuscript as follows.

*Page 18, Line 11:* '*...$\kappa$ was almost independent of $f_{Cl^-}$, partially due to the limited availability of non-refractory chloride. These distinctly different correlations between inorganics with mean $\kappa$ likely reflect the different formation mechanisms of these species. Sulfate formation occurs as the result of the photochemical process during the day, whereas diurnal variations in gas–particle partitioning are important for the mass concentration of nitrate or chloride (Aan de Brugh et al., 2012; Gong et al., 2013).*'

**R1C12: A relationship in figure 9 is likely but figure 8 shows little relationship (suggest dropping this and stating no relationship found). Even without Oct 22 the relationship looks weak, r$^2$ of 0.2 or so?**

**Response:** We appreciate the reviewer for the suggestion. We agree with the reviewer that the empirical correlation between $\kappa_{org}$ and $f_{WSOC}$ in Fig.8 is not that straightforward, as the correlation coefficient ($R$) is only 0.09. The two parameters tend to correlate more closely if not taking the Oct 22 data into account, and the $R$ value approximates 0.51. These results suggest that the bulk WSOC fraction could still influence water uptake by wildfire haze particles, considering the potential bias in $f_{WSOC}$ between 24h-averaged bulk PM$_{2.5}$ filter samples and real-time HTDMA measurements of 100 nm particles. In this sense, we would prefer to

show the correlation plot in the manuscript; while instead, we accept the reviewer's suggestion of stating that there was little relationship between $\kappa_{org}$ and $f_{WSOC}$ based on the analysis of ambient $PM_{2.5}$ filter samples.

The manuscript was revised as follows to clarify this point.

***Page 18, Line 23:*** '*... In general, $\kappa_{org}$ was insignificantly correlated with $f_{WSOC}$, especially in the data on October 22.*'

**Other comments on Presentation**

**R1C13: Overall the writing mechanics is reasonable and clear but needs further work for ACP standards. I will highlight a few passages with suggestions for better writing. I recommend further refinement with a fluent English writer to raise it to acceptable levels. Here are a few:**

**Response:** We acknowledge the reviewer for the critical comments, which are useful in improving the impact and quality of the manuscript. We have fully revised the manuscript according to all the helpful comments and suggestions, and the updated contents can be found as stated above and elsewhere in our responses to the second reviewer's comments. The revised manuscript was sent to a professional English editing company to improve the presentation quality.

**R1C14: Line 14 change to "not only in terms of hazards to human health"**

**Response:** We have revised it accordingly.

*Page 3, Line 14:* '...The recent equatorial Asian wildfire haze event in 2015 could rival the one in 1997 not only in terms of the hazards to human health but also the significant impacts on global climate ...'

**R1C15: Line 16 I'm not aware of any prizes offered for poor air quality and though I'm sure the region would vie for such a title I recommend keeping magazine-like statements out of the article.**

**Response:** We have deleted the whole sentence.

**R1C16: In Figure 5 I suggest either plotting d/do or kappa an not the rainbow colors of kappa that follows the same pattern as d/do that is plotted.**

**Response:** We have replotted Fig. 5 as below.

[Figure]

*Fig.5* *Time series of the volume-weighted mean particle diameter growth factor (GF) derived from HTDMA measurements (date format: Month/Day, 2015). The black dashed line stands for the temporal mean GF averaged over the entire observation period.*

**R1C17: The traces in figure 6 are somewhat difficult to distinguish."**

**Reference: Chemical characterization of fine particulate matter emitted by peat fires in Central Kalimantan, Indonesia, during the 2015 El Niño, Jayarathne et al. (2017) ACP.**

**Cloud condensation nuclei activity of fresh primary and aged biomass burning aerosol, G. J. Engelhart, Atmos. Chem. Phys., 12, 7285–7293, 2012, www.atmos-chemphys.net/12/7285/2012/doi:10.5194/acp-12-7285-2012**

**Response:** We appreciate the reviewer for the comment. We have updated Figure 6 as below, where the four OA factors (previously displayed in subfigure (d)) were separately plotted in panels (d-e) to distinguish each other more clearly.

[revised manuscript text omitted]

Kang, E., Root, M. J., Toohey, D. W., and Brune, W. H. (2007). Introducing the concept of Potential Aerosol Mass (PAM). Atmos. Chem. Phys., 7, 5727-5744.

Lambe, A. T., Ahern, A. T., Williams, L. R., Slowik, J. G., Wong, J. P. S., Abbatt, J. P. D., Brune, W. H., Ng, N. L., Wright, J. P., Croasdale, D. R., Worsnop, D. R., Davidovits, P., and Onasch, T. B. (2011a). Characterization of aerosol photooxidation flow reactors: heterogeneous oxidation, secondary organic aerosol formation and cloud condensation nuclei activity measurements. Atmos. Meas. Tech., 4, 445-461.

Lambe, A. T., Chhabra, P. S., Onasch, T. B., Brune, W. H., Hunter, J. F., Kroll, J. H., Cummings, M. J., Brogan, J. F., Parmar, Y., Worsnop, D. R., Kolb, C. E., and Davidovits, P. (2015). Effect of oxidant concentration, exposure time, and seed particles on secondary organic aerosol chemical composition and yield. Atmos. Chem. Phys., 15, 3063-3075.

Lambe, A. T., Onasch, T. B., Massoli, P., Croasdale, D. R., Wright, J. P., Ahern, A. T., Williams, L. R., Worsnop, D. R., Brune, W. H., and Davidovits, P. (2011b). Laboratory studies of the chemical composition and cloud condensation nuclei (CCN) activity of secondary organic aerosol (SOA) and oxidized primary organic aerosol (OPOA). Atmos. Chem. Phys., 11, 8913-8928.

Marlier, M. E., DeFries, R., Pennington, D., Nelson, E., Ordway, E. M., Lewis, J., Koplitz, S. N., and Mickley, L. J. (2015). Future fire emissions associated with projected land use change in Sumatra. Glob. Change Biol., 21, 345-362.

Miettinen, J., Hooijer, A., Wang, J., Shi, C., and Liew, S. C. (2012). Peatland degradation and conversion sequences and interrelations in Sumatra. Reg. Environ. Change, 12, 729-737.

Stockwell, C. E., Jayarathne, T., Cochrane, M. A., Ryan, K. C., Putra, E. I., Saharjo, B. H., Nurhayati, A. D., Albar, I., Blake, D. R., Simpson, I. J., Stone, E. A., Yokelson, R. J. (2016). Field measurements of trace gases and aerosols emitted by peat fires in Central Kalimantan, Indonesia, during the 2015 El Niño. Atmos. Chem. Phys., 16, 11711-11732.

Tang, I. N. and Munkelwitz, H. R. (1994). Water activities, densities, and refractive indices of aqueous sulfates and sodium nitrate droplets of atmospheric importance. J. Geophys. Res. Atmos., 99: 18,801-18,808.

**Anonymous Referee #2**

**General comments:**

**R2C0: This manuscript describes an aerosol dataset collected in Singapore during a period in October, 2015 when the region was impacted by high concentrations of smoke particles. Though I have some concerns with the analysis, I support eventual publication in ACP in part because the region and this particle source are understudied. But major revisions would be required first. In addition to the concerns identified below, the writing would need to be improved prior to publication. The mistakes are too numerous to identify in this review.**

**Response:** We appreciate the reviewer's supportive comments. We have organized the following responses to address the reviewer's concerns and revised the manuscript thoroughly. The revised manuscript was sent to a professional English editing company to correct any grammatical issues.

**Other comments**

**R2C1: A general concern I have with the manuscript is that it lacks a description of the meteorological and chemical setting that readers would need to understand the data. There is a brief mention of the location of the sampling site relative to a road and a petrochemical complex, but no discussion about whether and when those sources would be upwind. There is also no discussion of the typical transport time of the smoke prior to arrival at the sampling site. And to put the smoke-impacted measurements into context,**

**there needs to be some data and discussion of the typical concentration and composition at the site.**

**Response:**

We thank the reviewer for the helpful comments. In the revised manuscript, we added distributions of carbon emissions from wildfire (GFEDv4) as well as back trajectories of air masses arriving at Singapore. It took approximately 1~2 days for air mass transporting from the Southern part of Sumatra Island to Singapore, and 3~4 days for wildfire plumes from Central Kalimantan to arrive at Singapore.

The highway and petrochemical complex are located at south of the observation site. As shown in Fig. S1, the air masses were arriving at Singapore from Southeast. Although we do not have the local meteorological data, it is likely that the sampling site was influenced by these emission sources.

The above information on meteorological condition and chemical characteristics associated with our observations has been clarified in the revised manuscript as follows.

*Page 6, Line 7: '…located approximately 8 km south. In September−October 2015, the observation site encountered severe transboundary haze pollution that was caused by recurring Indonesian peatland fires, dominated by the smoldering combustion of underground organic-rich peat soils and mixed surface vegetation burning (Field et al., 2016; Page et al., 2009; Jayaranthe et al., 2017). Particles emitted from the wildfires had experienced for approximately 1−4 days atmospheric aging process before arriving at Singapore (Fig. S1).'*

**R2C2: Line 20 (and later): There is no discussion about the disconnect between the single size at which g was measured (100 nm) and the larger particles that dominate the composition measured by the ACSM. There is a mention that the number size distribution is dominated by particles in the 50 – 200 nm size range, but the number distribution is irrelevant. It seems likely that the diurnal variations in g would not be as important for particles near the mass median size of the distribution. So then the composition should be comparatively constant and the variation between it and g introduces scatter to the inferred hygroscopicity parameters. It also seems likely that much of the sulfate will be in >100 nm particles if it forms in cloud. So then it is questionable how much direct impact it has on g.**

**Response:** We appreciate the reviewer for pointing out this issue. We selected 100 nm particles for the present study, as the diameter is close to the mode diameter for number size distribution. We agree with the reviewer that the mode diameter for mass size distribution (~ 300 nm) was much larger than 100 nm, as shown in **Fig. S4**. This value is comparable to those in previous haze events in Singapore (Balasubramanian et al., 2003; See et al., 2006).

[Figure]

*Fig. S2* *Mean particle number (Num.) and volume (Vol.) size distributions over the entire wildfire haze observation period. Submicron particles with 200 nm < $D_p$ < 1 μm dominate the particle volume concentration, while 30−200 nm particles are the major contributor to the particle number concentration.*

In the revised manuscript, we have included the corresponding clarification.

***Page 13, Line 4:*** *'...whereas particles larger than 600 nm accounted for a minor fraction (less than 4.0 % on average; Fig. S4 and Fig. S5). This result suggests that in Singapore, the wildfire haze particles were predominantly contributed by submicron particles, in line with the corresponding chemical characteristics obtained in previous studies (Balasubramanian et al., 2003; See et al., 2006).'*

**R2C3: I am unfamiliar with the characteristics of PBOA. Is it reliably separated from BBOA? Could variation in fuel type or burning characteristics cause shifts in attribution between the two types? And what is known about the solubility of peat burning primary particles? I ask because it could influence the fraction of soluble species in solution at the 85% RH in the**

**HTDMA. These points should be added to the manuscript and not simply provided in a response.**

**Response:** We thank the reviewer for the comments. The details about the ME-2 analysis are available in our recent manuscript, which was submitted to ACP (Budisulistiorini et al., submitted). Briefly, the mass spectrum of PBOA factor is almost identical to the mass spectra for peat burning particles in laboratory data, as the initial guess was provided from our experimental data (Budisulistiorini et al., 2017). Mass spectra of peat burning particles, which are significantly different from those emitted from combustion of other types of biomass, are only slightly influenced by smoldering condition, as demonstrated by Kuwata et al. (2017). The contributions of PBOA and BBOA factors exhibit different patterns in time series, suggesting that they can be separated well (Budisulistiorini et al., submitted). We agree with the reviewer that a unique solution does not exist for such a factor analysis, causing ambiguity in interpretation of data. This point is clarified in the revised manuscript as follow.

*Page 7, Line 19: '... Four specific types of OA were identified: hydrocarbon-like OA (HOA), peat burning OA (PBOA), non-peat biomass burning OA (briefly BBOA), and oxygenated OA (OOA). The HOA factor was mainly contributed by primary sources, such as the combustion emissions from fossil fuel (e.g., related to traffic, shipping, and industrial use), excluding the influence of biomass burning. The PBOA and BBOA factors were well separated. Details about the ToF-ACSM measurements and data analysis are provided in Budisulistiorini et al. (submitted to Atmospheric Chemistry and Physics, February 2018).'*

The mean hygroscopicity parameter $\kappa$ of freshly emitted peat burning particles is approximately 0.04, which is limited by availability of water soluble organic compounds (WSOC/OC = 0.03) (Chen et al., 2017; Relevant details were also provided in our response to **R1C7**.). The result demonstrates that PBOA can practically be considered as nearly non-hygroscopic.

In the revised manuscript, we have included the corresponding clarification.

*Page 14, Line 16: '... and maximum values of 63.6 % and 79.9 %, respectively. The mean WSOC/OC result was significantly higher than that for fresh Indonesian peat burning particles emitted from the source region (i.e., 16 %; Jayaranthe et al., 2017), which were demonstrated to be generally water insoluble and thus nearly non-hygroscopic (Chen et al., 2017). This result suggests that the majority of organics in the wildfire haze particles were water soluble, implying the importance of secondary formation as well as the chemical transformation of organic particles during atmospheric transport.'*

**R2C4: Page 8, line 13: The OPS does not measure aerodynamic size.**

**Response:** The reviewer is right. We have revised it to optical size.

**R2C5: Page 9, Line 3: The only g distribution provided is the study average shown in Figure 1. The separation of particles into the three g categories implies that the distributions were generally multimodal. I don't doubt that, but it should be shown. Perhaps a few example distributions could be included in a supplemental document.**

**Response:** We thank the reviewer for the suggestion. We have added an example of normalized particle size distributions (in the period of 1−7 Oct 2015) after humidification at 85 % RH into the supplementary material. Accordingly, we have updated the manuscript as follows.

[Figure]

*Fig. S3* *An example (1−7 Oct 2015) of the HTDMA data at RH = 85% for $D_0$ = 100 nm particles. Temporal variation of normalized particle number size distributions (i.e., Norm. $dN/dlogD_p$) is shown, with every 12 hours interval.*

***Page 9, Line 8:*** *'... hygroscopic properties at RH = 85 % (temporal variation of the multimodal size distribution patterns are shown in Fig. S3 of the supplementary material), facilitating the analysis of heterogeneity of particle chemical composition.'*

**R2C6: Page 9, line 14: I think "More" would be a better word here. Highly to me implies much higher g than 1.27. Especially when those with g < 1.15 are "Nearly non-hygroscopic"; that's a large change in type over a small change in g.**

**Response:** We agree with the reviewer. In the revised manuscript, we have replaced the 'highly hygroscopic' by 'more hygroscopic'.

*Page 9, Line 19: '(3) More hygroscopic particles ($\kappa \geq 0.2$; $g \geq 1.27$): Aerosol particles contain inorganic salts as well as some more hygroscopic organic species such as ...'*

**R2C7: Page 10, line 9: Those are not salts.**

**Response:** We thank the reviewer for the comment. We have changed it into 'inorganic constitutes'.

*Page 10, Line 13: '...The subscript SNA represents the three major inorganic constituents of sulfate, nitrate, and ammonium;'*

**R2C8: Page 10, line 12: Is sea salt expected to be important at the site? Unless operated at a high temperature the ACSM would not see it.**

**Response:** Previous study on wildfire haze in Singapore demonstrated that sea salt (e.g., $Na^+$, $Cl^-$, approximately 1% ~ 3%) was much less abundant than organics (30 ~ 40%) or sulfate (~ 20%) (Balasubramanian et al., 2003; See et al., 2006). We added the following sentence in the main text to address the point.

*Page 10, Line 17: '...Other materials, such as sea salt and crustal elements, were demonstrated to be neglected because they are relatively scarce in submicron wildfire haze particles in Southeast Asia (Balasubramanian et al., 2003; Keywood et al., 2003; See et al., 2006; Stockwell et al., 2016).'*

**R2C9: Page 10, line 21: The second sentence in this paragraph needs to be rewritten.**

**Response:** We have rephrased it as below.

*Page 11, Line 1: 'The mass fraction is taken as the first-order approximation of the volume fraction, based on the hypothesis that the bulk particle density is similar to the densities of individual compounds when volume additivity is assumed ...'*

**R2C10: Page 11, line 15: I can't understand why the authors chose to assume that both BBOA and PBOA have kappas of 0.0. Even if they are low, why not use best estimates of the values instead of just arbitrarily setting them to 0. Because they are not actually completely non-hygroscopic, the result will be an erroneous sensitivity of kappa_OOA to the organic type fractions. This needs to be changed in the revision.**

**Response:** We thank the reviewer for the suggestion. In the revised manuscript, we used the values of $\kappa$, those were obtained in our previous laboratory study ($\kappa_{PBOA} = 0.04$, $\kappa_{BBOA} = 0.06$, Chen et al., 2017). The revised manuscript was updated accordingly, as detailed below.

*(Sect. 3.3) Page 11, Line 21: '... where $v_i$ stands for the volume fraction of component i in all the organics.*

[revised manuscript text omitted]

**R2C11: Page 12, line 21: I agree that most of wildfire haze mass will be submicron, but that can't be asserted based on the number concentration as**

**the authors do. At a minimum the supermicron volume fraction should be used for this conclusion.**

**Response:** We appreciate the reviewer for the helpful suggestion. We have calculated the total particle volume concentrations of $PM_1$ and $PM_{10}$, and corresponding volume fraction of $PM_1$ to $PM_{10}$ (as displayed in **Fig. S5**). The mean submicron volume fraction is higher than 0.7, revealing the dominance of submicron particles in total particle mass of wildfire haze particles in Singapore.

[Figure]

***Fig. S5*** *(a) Time series of total particle volume concentration (Vol. Conc.) for $PM_1$ and $PM_{10}$. (b) The corresponding volume fraction (Vol. frac.) of $PM_1$ to $PM_{10}$ calculated from the combined particle size distribution data observed with NanoScan SMPS and OPS. The red line represents the mean level (approximately 72.3 %) averaged over the entire wildfire haze period. This suggests the predominant role of submicron particles in total particle mass.*

Accordingly, we have updated the corresponding content in Sect. 4.1 of the manuscript as below.

*Page 13, Line 4: '…whereas particles larger than 600 nm accounted for a minor fraction (less than 4.0 % on average; Fig. S4 and Fig. S5). This result suggests that in Singapore, the wildfire haze particles were predominantly contributed by submicron particles, in line with the corresponding chemical characteristics obtained in previous studies (Balasubramanian et al., 2003; See et al., 2006).'*

**R2C12: Page 12, lines 25 and 26: dN/dlogDp is not the same as number concentration.**

**Response:** We appreciate the reviewer for the comment. We have updated the sentence as below.

*Page 13, Line 9: 'Figure 2b shows the mean diurnal cycle of particle number size distribution. The growth of ultrafine particles was typically observed in the afternoon. The $dN/dlogD_p$ higher than $1.5 \times 10^4$ $cm^{-3}$ was commonly observed in the 50–200 nm particle size range, while the $dN/dlogD_p$ of super micron particles seldom exceeded $1.0 \times 10^3$ $cm^{-3}$.'*

**R2C13: Page 13, line 1: The authors note that traffic emissions may be important but take no steps to correct for the impact in their analysis. Should periods when winds are from the road and when traffic is heavy be excluded? The novelty of this dataset is obviously the wildfire haze and anything local is an interference.**

**Response:** We appreciate the reviewer for pointing out this issue. We agree that HOA and EC are likely dominantly emitted from traffic or other types of industrial

combustion processes around the observation site. Contributions of these unavoidable locally emitted species were minimal (e.g., HOA/OA < 0.1; Budisulistiorini et al., submitted), suggesting that the dominant fraction of observed aerosol was from wildfire. As the contributions of locally emitted species are likely limited, no data point was excluded from the analysis.

We have included the corresponding clarification into the revised manuscript.

***Page 7, Line 19:*** *'Four specific types of OA were identified: hydrocarbon-like OA (HOA), peat burning OA (PBOA), non-peat biomass burning OA (briefly BBOA), and oxygenated OA (OOA). The HOA factor was mainly contributed by primary sources, such as the combustion emissions from fossil fuel (e.g., related to traffic, shipping, and industrial use), excluding the influence of biomass burning. The PBOA and BBOA factors were well separated. Details about the ToF-ACSM measurements and data analysis are provided in Budisulistiorini et al. (submitted to Atmospheric Chemistry and Physics, February 2018).'*

***Page 11, Line 21:*** *'... where $v_i$ stands for the volume fraction of component i in all the organics.*

*Water uptake by a mixed particle is largely driven by the relative abundance of more or less hygroscopic component, and it is more sensitive to uncertainties in the hygroscopicity of more hygroscopic compounds than that of less hygroscopic compounds (Gysel et al., 2007). Hydrocarbon (-like) OA is known to be almost non-hygroscopic, leading to the estimation that the $\kappa$ value of HOA is 0 (Gysel et al., 2007; Gunthe et al., 2009; Chang, et al., 2010).'*

**R2C14: Page 13, line 22: Add non-refractory in front of each instance of chloride.**

**Response:** We have updated the corresponding contents accordingly.

*Page 14, Line 6: 'The mass concentration of non-refractory chloride was almost negligible ...'*

*Page 18, Line 12: '... partially due to the limited availability of non-refractory chloride.'*

**R2C15: Page 14, line 2: Related to the above comment about the assumption that only the OOA contributed to hygroscopicity, the fraction of the organic mass that was water soluble was quite high (average 64%, max 80%). How frequently was this higher than the fraction of organics categorized as OOA?**

**Response:** The reviewer is right that the mean WSOC/OC ratio is approximately 63.6 % for the $PM_{2.5}$ filter samples collected on 8 specific days. Correspondingly, the mass fraction of WSOC in $PM_{2.5}$ ranges from 17.6–30.2 % (as displayed in Fig.4 and Fig.8), with a mean value of 25.3 %. On average, OOA accounts for 36.0 % of mass in $NR-PM_1$ ($f_{OOA}$) during the entire wildfire observation period. Although we need to be careful about comparing $PM_{2.5}$ and $NR-PM_1$ data because of the difference in their size ranges, this mean $f_{OOA}$ value is quite comparable to the mean fraction of WSOC in $PM_{2.5}$ mass.

**R2C16: Page 14, line 7: Is it surprising that there is much less K than $SO_4$?**

**Response:** Previous observation data at Singapore during similar wildfire events demonstrated that $SO_4^{2-}$ is much more abundant than $K^+$. For instance, the mass fraction of $SO_4^{2-}$ was found to be approximately 20%, while that for $K^+$ was around 3% (Balasubramanian et al., 2003; See et al., 2006). So, we do not consider that the observed concentration values of $K^+$ and $SO_4^{2-}$ are surprising.

**R2C17: Page 14, line 26: Somewhere in the paper there needs to be a conceptual explanation of what is responsible for the observed mix of HOA, PBOA, BBOA, and OOA. Do the authors believe that closer to the source measurements would show mostly PBOA and BBOA and that aging is responsible for the conversion to OOA? (the transport timescale issue mentioned above is relevant here). Is the daily variation thought to be mostly due to photochemical processing or could it also be daily variation in fire characteristics? And is it thought that the changes are due to processing of the existing particles or production of SOA from gas phase emissions from the fires? The one example size distribution time series presented does not seem to indicate particles are growing, but rather the size is about constant and only the concentration changes. Do the time series on other days look like that as well? And if so, what is responsible?**

**Response:** We thank the reviewer for the comment. We think that PBOA and BBOA factors will dominate the contributions to OA at a vicinity of the source region. The OOA factor, which is much more oxygenated than any types of POAs, should be associated with both aging of primary organic particles and secondary formation processes (Stockwell et al., 2016). As the observation is located far from the source regions (1~4 days), diurnal variation in fire characteristics (if any)

would not directly influence that of our observation data directly. Considering these conditions, we think that the diurnal variation of OOA factor is mainly caused by *in-situ* photochemical processing of primary particles and/or secondary formation from VOC precursors, rather than by the daily variation in fire characteristics.

The size distribution displayed in Fig. 2(b) actually demonstrates the mean diurnal variation of particle number size distribution. Growth of ultrafine particles during afternoon period is evident in the figure. It is also evident in data on 2 October 2015, which is shown in the following figure.

[Figure]

We added the following sentence to the revised manuscript to stress the point.

***Page 13, Line 9:*** *'Figure 2b shows the mean diurnal cycle of particle number size distribution. The growth of ultrafine particles was typically observed in the afternoon.'*

**R2C18: Page 17, line 15: NO₃ was a small contributor to the overall mass. It seems pointless to note that variations in that small contribution had little effect on kappa.**

**Response:** We appreciate the reviewer for the comment. We have updated the corresponding description in a following way.

***Page 18, Line 9:*** *'... There was no clear correlation between $\kappa$ and $f_{NO_3^-}$, implying that the small amount of nitrate had an insignificant contribution to the variability in $\kappa$ of wildfire haze particles.'*

**R2C19: Figure 3: I would like to be able to see whether organic mass and sulfate mass are correlated. But with the use of the same y-scale for both in (a) that is not possible.**

**Response:** We have provided the correlation plots of organics mass concentration vs. sulfate mass concentration (both with and without the data for the extremely pollution episode in the evening of 19 October 2015 till the noon time of the next day) into supplementary material. The corresponding correlation coefficients are $R$ = 0.49 (without) and 0.62 (with), respectively, suggesting that both of the two chemical species could be from the peatland fires in Indonesia.

[Figure]

***Fig. S6*** *Correlation between mass concentration of organics and that of sulfate (R = 0.49), without the data measured during the extremely pollution episode in the evening of 19 October 2015 till the noon time of 20 October 2015.*

[Figure]

***Fig. S7*** *Correlation between mass concentration of organics and that of sulfate (R = 0.62) during the entire wildfire haze observation period.*

**R2C20: Figure 6: Specify date.**

**Response:** Diurnal variations in this figure are averaged over the entire observation period (10−24 October 2015). We have added the time period into the caption of Figure 6 for clarification.

**Caption of Figure 6**: '*Fig.6 Temporal averaged (10−24 October 2015) diurnal variations of …*'

**R2C21: Figure 9: Clarify what "averaged diurnal" means.**

**Response:** "averaged diurnal" represents that the diurnal results for each day were further averaged over the period when both the HTDMA and ToF-ACSM data were available (i.e., 10−24 October 2015). We have changed it into 'mean diurnal' and added the corresponding information to the caption of Fig.9 for clarification.

**Caption of Figure 9**: '*Relationships between the mean diurnal $\kappa_{org}$ results vs. (a) $f_{44}$ and (b) $f_{44/43}$ in NR-PM$_1$ haze particles. Mean diurnal here represents that the diurnal results for each day were further averaged over the overlapping observation period, i.e., with both HTDMA and ToF-ACSM measurements taken from 10−24 October 2015.*'

[revised manuscript text omitted]

**S1. Back trajectories and carbon emission from wildfires**

[Figure]

**Fig. S1** Back trajectories of air masses arriving at Singapore and monthly carbon emission from wildfires during the observation period of HTDMA measurements. The transport time of wildfire plumes is approximately 1−2 days from South Sumatra, and 3−4 days from Central Kalimantan arriving at Singapore.

The back trajectory was calculated using the NOAA HYSPLIT model at 500 m (Kalnay et al., 1996). The altitude of the trajectories was constrained as iso-sigma. Carbon emission data was from the Global Fire Emissions Database (GFEDv4, https://daac.ornl.gov/VEGETATION/guides/fire_emissions_v4.html).

**S2. HTDMA calibration results with 150 nm ammonium sulfate particles**

[Figure]

**Fig. S2** Comparison of particle diameter growth factor (GF) results derived from the HTDMA calibration data for 150 nm dry ammonium sulfate (AS) particles, and experimental data (without parameterization) obtained by Tang and Munkelwitz (1994).

**S3. An example of normalized particle number size distributions of 100 nm particles after humidification at 85 % RH measured using the HTDMA system**

[Figure]

**Fig. S3** An example (1−7 Oct 2015) of the HTDMA data at RH = 85% for $D_0$ = 100 nm particles. Temporal variation of normalized particle number size distributions (i.e., Norm. dN/dlogD$_p$) is shown, with every 12 hours interval.

**S4. Mean particle number and volume size distributions**

[Figure]

**Fig. S4** Mean particle number (Num.) and volume (Vol.) size distributions over the entire wildfire haze observation period. Submicron particles with 200 nm $< D_p < 1$ μm dominate the particle volume concentration, while 30−200 nm particles are the major contributor to the particle number concentration.

**S5. Time series of total particle volume concentrations for PM$_1$ and PM$_{10}$ measured during the wildfire haze periods**

[Figure]

**Fig. S5** (a) Time series of total particle volume concentrations (Vol. Conc.) for PM$_1$ and PM$_{10}$. (b) The corresponding volume fraction (Vol. frac.) of PM$_1$ to PM$_{10}$ calculated from the combined particle size distribution. The data from the NanoScan SMPS and OPS were used for the analysis. The red line represents the mean level (approximately 72.3 %) averaged over the entire observation period. This result suggests the predominant role of submicron particles in total particle mass.

**S6. Correlations between the mass concentration of organics and that of sulfate**

[Figure]

**Fig. S6** Correlation between mass concentration of organics and that of sulfate ($R = 0.49$), without the data measured during the extremely pollution episode in the evening of 19 October 2015 till the noon time of 20 October 2015.

[Figure]

**Fig. S7** Correlation between mass concentration of organics and that of sulfate ($R = 0.62$) during the entire wildfire haze observation period.

---

## Author Response (AR2)

Dear Editor,

We would like to thank both referees for their supportive comments. The specific concerns raised by the reviewer #2 have been fully considered upon manuscript revision. A point-by-point response and an accordingly updated manuscript have been uploaded.

In the following, original reviewer comments, our response, and updates on the revised manuscript are shown in **bold**, normal, and *italic*, respectively.

Kind Regards,

Jing Chen, Mikinori Kuwata

**Anonymous Referee #2**

**General comments:**

**R2C0: The authors satisfactorily responded to most of my concerns. The two exceptions are noted below.**

**Response:** We appreciate the reviewer's comments. We have organized the following responses to address the specific concerns and revised the manuscript thoroughly.

**Specific comments:**

**R2C1: The changes made in response to my concern about the importance of the disconnect between the ACSM composition and 100 nm growth factor are not adequate. The text should describe the impact of the difference on the results and on the uncertainty in those results.**

**Response:** We would like to thank the reviewer for the comment. We admit that the possible difference in the chemical composition between 100 nm particles and all the non-refractory submicron particles (NR-PM$_1$) could have induced some ambiguity for the comparison of the data measured by HTDMA and the ToF-ACSM. Similar combinations of the ACSM data and HTDMA/CCN data are commonly used for detailed discussion in previous studies (e.g., Whitehead et al., 2016). These studies demonstrated that the ACSM data are helpful in qualitatively connecting chemical composition and hygroscopic growth, although the difference in particle size range might occasionally be important for quantitative analysis.

We have added the corresponding clarification into the revised manuscript.

*Page 11, Line 16: '... and that there was no significant difference between BC and EC. It should be noted that some difference in chemical composition could have existed between particles quantified by the ToF-ACSM (NR-PM$_1$) and HTDMA (100 nm). Although the data obtained by these two techniques are frequently combined for detailed analysis (e.g., Whitehead et al., 2016), the difference in particle size range along with the particle mixing state (Whitehead et al., 2014) can be a source of uncertainty for the following discussion in this work.'*

**R2C1: I'm still not satisfied with the explanation provided about the observed diurnal patterns. In their response the authors argue that diurnal changes in fire characteristics are not responsible because of the 1-4 day transport to the sampling location. But continuing with the same logic, shouldn't (almost) every sampled particle have gone through a similar daily photochemical cycle at least twice? And if so, how could that be reconciled with the jump in growth factor from say 1.15 to 1.45 observed on many of the days in Figure 5? In other words, if photochemical processing results in such a dramatic increase in hygroscopicity in one day, why aren't there 100 nm particles that went through such processing the day before (and maybe the day before that) and start off with such high hygroscopicity?**

**Response:** We appreciate the reviewer's comments. The diurnal variation in the mean growth factor could be attributed to the combined influences of long-range transport from Indonesian peatland fires (including fresh PBOA and BBOA, aged and/or secondary formed aerosols, Stockwell et al., 2016), local emissions (such as vehicles-related HOA, black carbon, SO$_2$ and NO$_x$ for *in-situ* secondary inorganics formation, and SOA precursors including anthropogenic and biogenic VOCs), and development of mixing layer (Krautstrunk et al., 2000; Geiß et al., 2017).

At the current moment, we are unaware of any evidence for cyclic diurnal variations about Indonesian peatland fires. The main focus of our present manuscript is about the relationship between chemical composition and hygroscopic properties of haze particles originating from peatland fires, rather than a detailed analysis about the cause of the temporal variation. Brief statements for possible factors, those could influence the variation of hygroscopic growth of the haze particles, were added to the revised manuscript to address the reviewer's concern. A detailed analysis of temporal variation of the chemical composition during the same observation period is provided in our different manuscript (Budisulistiorini et al., 2018).

We have added the following sentences into the revised manuscript to stress the point.

*Page 7, Line 23:* *'The PBOA and BBOA factors were attributed to the long-range transport from Indonesian peatland fires, and the two OA factors were well separated.'*

[revised manuscript text omitted]